# THE GEOMETRY OF TOKENS IN INTERNAL REPRESENTATIONS OF LARGE LANGUAGE MODELS

## ABSTRACT

We investigate the relationship between the geometry of token embeddings and their role in next token prediction within transformer models. Toward this goal, previous studies have utilized metrics such as intrinsic dimension and neighborhood overlap to probe the geometry of internal representations, where prompts are summarized as a single point in representation space. We expand single points to point clouds by investigating how models geometrically distribute tokens in their internal representations. We measure the intrinsic dimension, neighborhood overlap, and cosine similarity on these point clouds for a large number of prompts. To validate our approach, we compare these metrics to a dataset where the tokens are shuffled, which disrupts the syntactic and semantic structure. Our analysis reveals a correlation between the geometric properties of token embeddings and the cross-entropy loss of next token predictions, implying that prompts with higher loss values have tokens represented in higher-dimensional spaces.

## 1   INTRODUCTION

In the context of interpretability of transformer models, a set of analytic approaches have been developed with the goal of modeling transformer architectures as dynamical systems of particles (Vuckovic et al., 2020; Geshkovski et al., 2024b; Cowsik et al., 2024). In this perspective, the transformers are viewed as evolving a mean-field interacting particle system where the evolution of tokens across layers is controlled by their empirical measure[1] (Agrachev & Letrouit, 2024). Under a set of assumptions such as time-independent weights, this interpretation is used to show that tokens tend to cluster in the later layers (Geshkovski et al., 2023). This clustering behavior can be associated with the empirically observed rank collapse phenomenon in transformer models (Anagnostidis et al., 2022; Shi et al., 2022; Wu et al., 2023; He et al., 2023; Wu et al., 2024).

An important insight from Geshkovski et al. (2024b) in the context of next token prediction is that the output measure of tokens encodes the probability distribution of the next token, and its clustering indicates a small number of possible outcomes. A complementary perspective to the evolution of token representations across layers can be gained by studying the latent predictions of transformer models (Belrose et al., 2023) from the perspective of iterative inference Jastrzebski et al. (2018) which indicates that the probabilities of the next tokens are incrementally updated layer by layer. The work by nostalgebraist (2020) suggests that causal LLMs appear to develop a reasonably accurate prediction regarding the next token in the middle layers, with subsequent layers refining these predictions. This means we should expect the empirical measures of the internal layers to reflect this trend, i.e. a rapid change of the empirical measure in the early layers and a more refined change towards the later layers. Since the latent predictions are obtained by unembedding the residual stream (Elhage et al., 2021), and our methods understand the geometric properties of the residual stream, we can expect the statistical properties (eg. entropy) of the latent prediction probabilities to be encoded in the geometry of the internal representations of the tokens.

In this work, we combine these viewpoints to examine the empirical measure of the internal layers from a geometric perspective. To observationally probe the empirical measure, we draw inspiration from previous works using intrinsic dimension and neighbourhood overlap to study the geometry

---

[1]In this context, the empirical measure and the output measure are used to characterize the distribution of tokens in the internal layers and the output layer respetively

of internal representations Ansuini et al. (2019); Doimo et al. (2020a); Pope et al. (2021); Valeriani et al. (2023); Cheng et al. (2023; 2024); Cheng & Antonello (2024). In these works, an important difference is that point clouds are built as a collection of prompts represented as a single point (the last token), not from the full sequence of tokens in a prompt, thereby lacking a direct link to the empirical measure. Additionally, we also calculate cosine similarity as a general probe of pairwise relations among tokens.

To test how the geometric properties of token representations change as a function of the model's internal dynamics, we probe it in a regime where the syntactic and semantic structures of the prompts are disrupted through systematic token shuffling. Our analysis achieves these main results:

- **Token-Level Intrinsic Dimension and Cosine Similarity**: We observe that the intrinsic dimension (ID) of token representations generally exhibits a peak, whose height increases with the degree of token shuffling. This peak is located at early to middle layers of the models. On the other hand, cosine similarity among tokens increases with shuffling, indicating increased alignment of token vectors.

- **Neighborhood Overlap Consistency**: The neighborhood overlap (NO) metric shows that token relationships around the ID peak become less consistent as the amount of shuffling increases. This highlights that structured data retains more coherent token neighborhoods through the model layers compared to shuffled data.

- **Correlation with Model Loss**: We find a statistical relation between the geometry of tokens and the probability distribution of the next token: the intrinsic dimension of the token representations across hidden layers is correlated to the average cross-entropy loss of the next token probability distribution for a given prompt. This suggests that prompts with a higher cross-entropy loss have token representations lying in higher dimensional manifolds.

## 2 RELATED WORK

**Broader context of mechanistic interpretability in transformers.** Mechanistic interpretability in transformers explores how transformer models encode and utilize information, focusing on semantic and algorithmic interpretations. Semantic interpretation investigates what latent properties are learned by models and how individual neurons may code for specific concepts (Hamrick & Mohamed, 2020). Structural probing (Rogers et al., 2020; Belinkov, 2022; Belinkov et al., 2020) and dictionary learning (Lewicki & Sejnowski, 2000; Lee et al., 2006; Faruqui et al., 2015) offer insights into how features are represented and reconstructed in transformer architectures. Relevant to this work, is the approach of the logit lens (nostalgebraist, 2020). This method offers insight into a model's predictive process by applying the final classification layer, which converts the residual stream activation into logits/vocabulary space, to intermediate residual stream activations. This reveals how prediction confidence evolves throughout the computational stages. This is feasible because transformers typically construct their predictions iteratively across layers (Geva et al., 2022). Building on this concept, the tuned lens (Belrose et al., 2023) employs affine probes to translate internal representations into probability distributions over the vocabulary. Similarly, the Future Lens (Pal et al., 2023) examines how individual representations encode information about forthcoming tokens.

**Analytic Approaches to Transformer Models** Recent analytical works (Geshkovski et al., 2023; Cowsik et al., 2024) indicate that analyzing geometric properties of token representations and their dynamics can offer meaningful insights into how transformers function. Geshkovski et al. (2023) introduced the novel perspective of viewing the evolution of tokens in the transformer layers as particles in a dynamical system. They predict clustering behavior in transformer models in a simplified setting which was later extended to include causally masked attention (Karagodin et al., 2024). (Cowsik et al., 2024) adopts the above perspective and examines particle geometry in the presence of MLP layers. This perspective not only offers insights into the geometric dynamics of tokens but also addresses the trainability of transformers based on initialization hyperparameters, including the strength of attentional and MLP residual connections. Further studies (Geshkovski et al., 2024c;a) theoretically investigate the expressive power of transformers as maps from arbitrary input measures to output measures and prove the appearance of dynamic metastability, i.e. the particles cluster in the infinite time limit but they resemble a configuration of several clusters for a long period of time. This behavior aligns more closely with practical observations than the clustering dynamics. This

analytical framework highlights the significance of studying the distribution of the internal representations of the tokens (referred to as the *empirical measure*) by i) suggesting a relation between the empirical measure to the next token prediction loss (Geshkovski et al., 2024b) ii) understanding the role of the empirical measure in governing the token dynamics (Agrachev & Letrouit, 2024).

**Geometric Approaches to Transformer Models.** The manifold hypothesis posits that real-world high-dimensional data often lie on or near a lower-dimensional manifold within the high-dimensional space (Goodfellow et al., 2016). The dimension of this approximating manifold is usually named the *intrinsic dimension* of the data. Several studies have demonstrated that the intrinsic dimension of data representations in deep networks shows a remarkable dynamic range, characterized by distinct phases of expansion and contraction (Ansuini et al., 2019; Doimo et al., 2020a; Pope et al., 2021). Data manifolds created by internal representations in deep networks have been also explored from the perspective of neuroscience and statistical mechanics (Chung et al., 2018; Cohen et al., 2020). In LLMs, a geometric analysis of representations has uncovered a rich set of phenomena. Geometric properties, such as intrinsic dimension and the composition of nearest neighbors, evolve throughout the network's sequence of internal layers. These changes mark distinct phases in the model's operation, signal the localization of semantic information (Valeriani et al., 2023; Cheng et al., 2023). While the aforementioned works analyze internal representations in linguistic processing, the geometry of context embeddings has been linked to language statistics (Zhao et al., 2024) and used to highlight differences between real and artificial data (Tulchinskii et al., 2023).

## 3 METHOD

Transformer models take as input a sequence of vectors embedded in $d$-dimensions of varying length $N$, $\{x_i\}_{i \in [N]} \in \mathbb{R}^{d \times N}$. Each element of the sequence is called a *token*, while the entire sequence is a *prompt*. A transformer is then a sequence of maps:

$$\{x_i(1)\}_{i \in [N]} \rightarrow \{x_i(2)\}_{i \in [N]} \cdots \rightarrow \{x_i(N_{\text{layers}})\}_{i \in [N]}, \tag{1}$$

where $x_i(\ell) \in \mathbb{R}^{d \times N}$ represents the $i$-th token at layer $\ell$, $N_{\text{layers}}$ the total number of model layers and $N$ is the number of tokens.

In transformer models, prompts can vary based on the specific application, representing protein sequences, image pixels, or sentences. In this study, we focus on causal language models and use sentences as our input prompts, though the technique can be extended to other input types as well. The prompt size can significantly vary depending on the dataset considered: sentences can be $\mathcal{O}(10)$ - $\mathcal{O}(1000)$ tokens long. Given that our goal is to study and interpret the geometrical behavior at the token level across model layers, we select prompts with a sufficient number of tokens, i.e. $N \geq 1024$ tokens, to ensure reliable estimates of our observables.

**Empirical measure.** Given $n$ points at positions $x_1, \ldots, x_n \in \mathbb{R}^d$ (a point cloud), their empirical measure is the probability measure $\mu = \frac{1}{n} \sum_{j=1}^{n} \delta_{x_j}$, i.e., the empirical measure encodes the distribution of points in the embedding space. In the context of transformers (Geshkovski et al., 2024b), the empirical measure characterizes the distribution of the tokens at each layer of the sequence 1. The empirical measure for the last layer is the *output* measure. The dynamical evolution of tokens in this framework, as described by Equation (1) in Agrachev & Letrouit (2024), indicates that the change in the token representation of token $i$ is controlled by a layer-dependent kernel $K_\ell$ and depends purely on the current token representation $x_i(\ell)$ and the empirical measure[2]. To probe the empirical measure across layers, we use cosine similarity, intrinsic dimension, and neighborhood overlap, as defined below.

**Intrinsic Dimension.** A substantial body of literature focuses on developing precise estimators for the intrinsic dimension of manifolds (Facco et al., 2017). In particular, nearest-neighbors-based algorithms are robust to high dimensionality and capture the non-linear structure of the manifold. In addition, it has been argued that a scale-sensitive algorithm can provide a stable estimation of the dimension, as it allows us to find the proper range of scale where the dimension is constant.

GRIDE (Denti et al., 2021) is a likelihood-based ID estimator that estimates the intrinsic dimension $\hat{d}(n_1, n_2)$ using the ratios $\dot{\mu} = \mu_{i,n_1,n_2} = \frac{r_{i,n_2}}{r_{i,n_1}}$, where $r_{i,k}$ is the Euclidean distance between point

---

[2] the dynamics of a token $i$ depends on the position of all the tokens $x_j(\ell)$ but not on their labels, which is an assumption in the mean-field interacting particle framework.

$i$ and its $k$-th nearest neighbour and $1 \leq n_1 < n_2$. Under the assumption of local uniform density, the distribution of $\mu_{i,n_1,n_2}$ is given by,

$$f_{\mu_{i,n_1,n_2}}(\dot{\mu}, d) = \frac{d\left(\dot{\mu}^d - 1\right)^{n_2-n_1-1}}{\dot{\mu}^{(n_2-1)d+1} B\left(n_2 - n_1, n_1\right)}, \quad \dot{\mu} > 1 \tag{2}$$

where $B(\cdot, \cdot)$ is the beta function. The ID estimate $\hat{d}(n_1, n_2)$ is obtained by maximizing the above likelihood with respect to $d$ assuming that the ratios $\mu_{i,n_1,n_2}$ are independent for different points. The conventional choice for the GRIDE algorithm is to set $n_2 = 2n_1$ and examine the variation of $\hat{d}$ for $n_2 \in \{2, 4, 8..\}$, where the parameter $n_2$ is known as the range scaling parameter. In this work, we mainly work with range scaling $= 2$ unless explicitly mentioned.

In the following analysis, we exploit the result for the TWO-NN estimator (Facco et al., 2017):

$$\hat{d}_{\text{TWO-NN}} = \frac{N - 1}{\sum_i^N \log\left(\mu_{i,1,2}\right)}, \tag{3}$$

which relates ID to the generic ratios $\mu_{i,1,2}$ thereby implying that there is an inverse relation between the dimension estimate and the generic ratios $(\mu_{i,1,2})$. Intuitively, we can expect a higher dimensional estimate to imply a lower $(\mu_{i,1,2})$ on average. [3]

**Neighborhood Overlap.** The neighborhood overlap $\chi_k^{l,m}$ was introduced in Doimo et al. (2020b) to measure similarity between representations in different layers $\ell, m$ at a given scale $k$. Given the representations of $N$ tokens in layers $\ell$ and $m$, we can define $\chi_k^{\ell,m}$ as

$$\chi_k^{\ell,m} = \frac{1}{N} \sum_i \frac{1}{k} \sum_{j \in \mathcal{N}_k^\ell(i)} \mathbb{I}\left(j \in \mathcal{N}_k^m(i)\right) \tag{4}$$

where $\mathcal{N}_k^\ell(i)$ is the set of $k$-nearest neighbors of a token $i$ in layer $\ell$. Intuitively, it measures the average number of shared $k$-nearest neighbors in layers $\ell, m$. In our context, we set $m = \ell + 1$, i.e. we calculate the neighborhood overlap between adjacent layers. By doing so, we measure the change in pairwise relations among tokens between successive layers.

## 4 EXPERIMENTS

### 4.1 MODELS AND DATASETS

**Models.** In this work, we analyze 3 different pre-trained decoder-only LLMs of similar dimensions: Llama 3 8B (Meta, 2024), Mistral 7B (Jiang et al., 2023), Pythia 6.9B(Biderman et al., 2023), each of them featuring 32 hidden layers. For brevity, we call them LLAMA, MISTRAL, and PYTHIA from now on. In the plots, layer 0 represents the embedding layer, with the hidden layers starting from layer 1. We extract internal representations of these models using a public library[4], where the token representations correspond to the representations in the residual stream (Elhage et al., 2021) after one attention and one MLP update.

**Datasets.** As a dataset representative of text in an extensive way, we use the Pile dataset which comprehends text from 22 different sources (Gao et al., 2020). We remark that Pile was used to train PYTHIA, a key consideration for the results discussed in Section 4.3. For computational reasons, we opted for the reduced size version Pile-10K (Nanda, 2022). We further filter only prompts of sequence length $N \geq 1024$ according to the tokenization schemes of all the above models. This choice ensures a reliable ID estimate. This results in 2244 prompts after filtering. We truncate the prompts by keeping the first $N = 1024$ tokens to eliminate the length-induced bias of our ID estimates if it were to be present.

---

[3]Note that the main assumption of the estimator is local homogeneity (Poisson distributed points within the local neighborhood of the point), which is generally true on a wide range of datasets.

[4]The library can be found in https://huggingface.co/docs/transformers/en/index

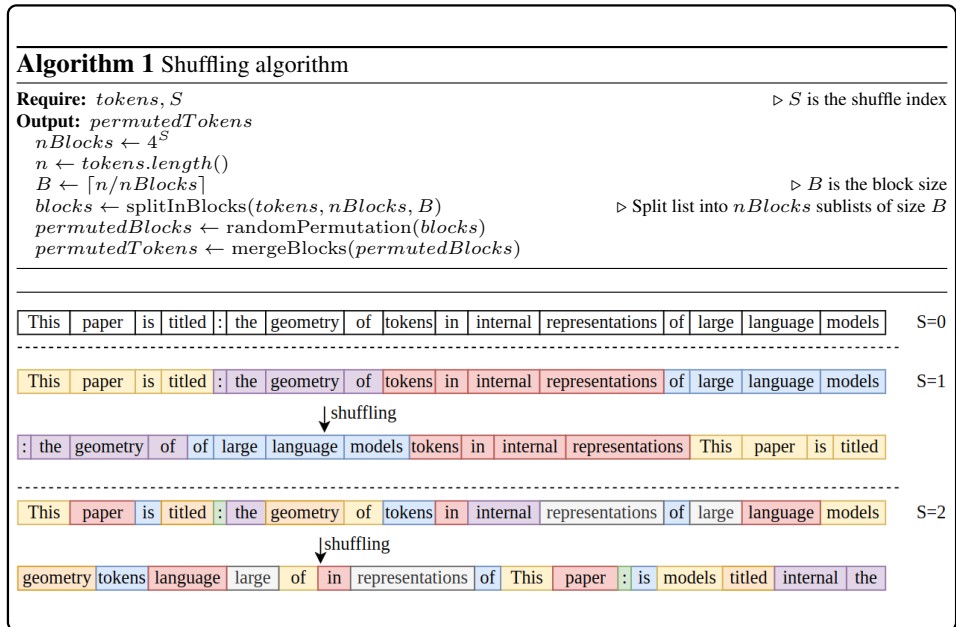

Figure 1: **The shuffling algorithm with an example.** Top Panel: Algorithmic description of the shuffling procedure described in Section 4.2. Bottom Panel: An example of the shuffling algorithm using $N = 16$ tokens. The first row ($S = 0$) corresponds to the unshuffled sequence. When $S = 1$, the tokens are split into $4^1$ blocks first and then, the blocks are shuffled. The last row $S = 2$ shows the fully shuffled case where the tokens are randomly permuted.

## 4.2 THE GEOMETRY OF SHUFFLED AND UNSHUFFLED PROMPTS

Evaluating geometric observables at the token level directly probes the model's internal dynamics. As a way of quantifying geometric changes, we compare in-distribution data to various levels of token shuffling. By progressively disrupting the syntactic and semantic structure while preserving unigram frequency distribution, we observe the incremental effects on our observables across layers.

**Shuffling method.** We define the shuffling of tokens in the following way: given a prompt with $N$ tokens, $X = \{x_i\}_{i \in [N]}$, we split the sequence into $nBlocks$ blocks of size $B$ such that $nBlocks \times B = N$ and take one random permutation of the blocks, as schematically presented in Figure 1. Note that the shuffle index for the fully shuffled case ($\hat{S}$) corresponds to the value of $S$ when the number of tokens $N = 4^{\hat{S}}$. In Figure 1, we have $\hat{S} = 2$ since we consider 16 tokens, whereas in the experiments, we have $\hat{S} = 5$ because we have $1024 = 4^5$ tokens.

In most of our experiments, we show two main results: i) the effect of various degrees of shuffling on our metrics for a single, random prompt and ii) the qualitative behavior of the unshuffled and the fully shuffled prompts on average. For the former observable, we consider the the $3218^{\text{th}}$ prompt from the Pile-10K dataset, with the Pile set name: *ArXiv*. This prompt is shuffled to six different levels labeled by ($S = 0, 1, \ldots, 5$) where the shuffle index $S$ quantifies the degree of shuffling: $S = 0$ represents the unshuffled state, while $S = 5$ corresponds to the fully shuffled case. We study the representations of this prompt using representations from LLAMA. [5]. For the average behavior, we find the averages of the geometric quantities (cosine similarity, ID and NO) over 2244 prompts.

---

[5]The qualitative behavior discussed in this section holds in general for other prompts and models. We show this in the case of intrinsic dimension by looking at the ID profile of other prompts using LLAMA (Figure 11) and the ID profile of $3218^{\text{th}}$ prompt in other models (Figure 12)

### 4.2.1 COSINE SIMILARITY

As a first step into investigating the geometry of internal representations at the token level, we compute the cosine similarity among tokens for each layer. In Figure 2, we show the average cosine similarity for different levels of shuffling as a function of model layers on a single prompt (Left Panel) and for the average over all prompts (Right Panel) for the LLAMA model. We can see that the cosine similarity increases with increasing shuffling and increasing layers. This implies that tokens are distributed along the same direction towards the last layers. For the structured prompts, the average cosine similarity is closer to zero, indicating that their directions are more orthogonal.

These results seem related to earlier works: Ethayarajh (2019) computes the average cosine similarity of randomly sampled words from BERT, GPT-2 and ELMo models across layers, finding high cosine similarity, in agreement with our shuffled case. In Liang et al. (2022) average cosine similarity was also computed on pre-trained text transformers, finding an average value of $\approx 0.5$ in the last layer.

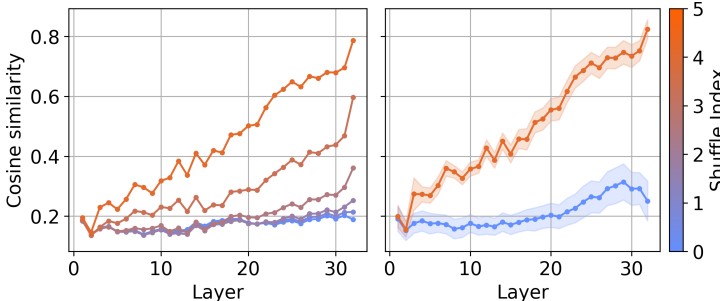

Figure 2: **Average Cosine Similarity.** Left Panel: average cosine similarity among tokens for a single prompt as a function of model layers. Right Panel: average cosine similarity averaged over 2244 prompts as a function of layers for the full shuffle ($S = 5$) and the structured case ($S = 0$). The color bar indicates the shuffle index $S$. The shaded regions indicate the standard deviation from the mean. All curves have been calculated for the LLAMA model.

### 4.2.2 INTRINSIC DIMENSION

Next, we examine the intrinsic dimension (ID) profile of tokens as a function of layers. Figure 3 displays the ID calculated for a range scaling of 2 for LLAMA. The Left Panel shows the ID profile of a single prompt at various levels of shuffling, while the Right Panel presents the average ID across 2244 prompts for both fully shuffled and structured cases. In all scenarios, we observe a peak in ID in the early to middle layers. Additionally, the height of this peak increases with the degree of shuffling, indicating a correlation between the two. Previous work focusing on studies of the geometry of internal representations at the prompt-level have investigated similar metrics. We devote a detailed comparison to Appendix A.

**Distribution of tokens at the ID peak.** We consider the relation in equation 3 between ID and the generic ratios $r_{i,2}/r_{i,1}$, i.e. the ratio of the distance of the second neighbor over the one of the first neighbor to the $i$-th point. According to equation 3, as ID grows we expect the ratio to tend to unity on average, implying that the first two nearest neighbors are roughly at equal distance from the reference token. On the other hand, if ID decreases we expect the two nearest neighbors to be at more varying distances. Therefore, a higher ID at the peak means that the nearest neighbors tend to be more equidistant for the shuffled prompts.

Furthermore, we can look at the angular distribution of nearest neighbors of a given point. Hence, we compute the cosine similarity between $x_{i,1} - x_i$ and $x_{i,2} - x_i$ for each token $i$ to determine the distribution of the angle formed by the first two nearest neighbors centered at token $i$.[6] We visualize this in Figure 4. In the left panel, we show the histogram of the angles between $x_{i,1} - x_i$

---

[6]In this paragraph, $x_{i,k}$ denotes the $k^{\text{th}}$ nearest token to token $i$ and $r_{i,k}$ is the distance between token $i$ and its $k^{\text{th}}$ nearest token.

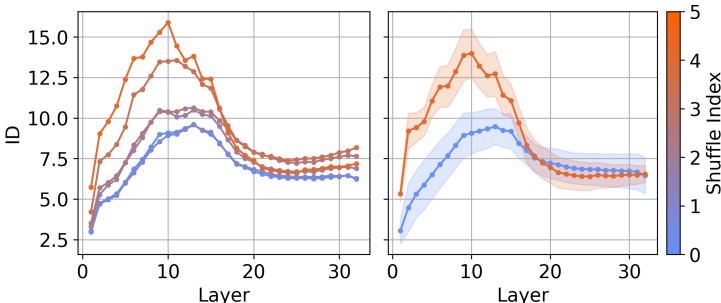

Figure 3: **Intrinsic Dimension.** Left Panel: intrinsic dimension for a single random prompt as a function of model layers. Right Panel: intrinsic dimension averaged over 2244 prompts as a function of layers for the full shuffle ($S = 5$) and the structured case ($S = 0$). The shaded regions indicate the standard deviation from the mean. The color bar indicates the shuffle index $S$. All curves have been calculated for the LLAMA model.

and $x_{i,2} - x_i$ for each token $i$ of a random prompt, at layer 10 of LLAMA, i.e. around the ID peak. In the right panel, we show the histogram of means over 2244 prompts for the full shuffle and structured cases. The distributions of mean angles differ between the two cases, with the mean angle between the nearest tokens being closer to 60 degrees for shuffled prompts. Combined with the earlier observation that the ratio $r_{i,2}/r_{i,1}$ is closer to unity, this suggests that the triangle formed by $x_i$, $x_{i,1}$ and $x_{i,2}$ is more equilateral in the full shuffle case at the ID peak. These findings suggest a distinguishable arrangement of tokens for shuffled prompts that deserve further investigation.

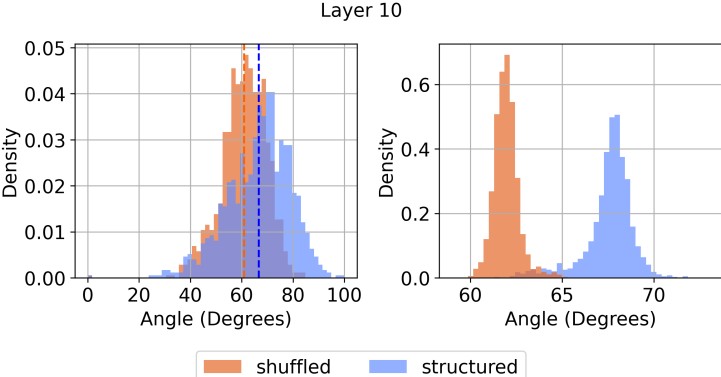

Figure 4: **Angle distribution between nearest neighbors.** Left Panel: histogram of the angles between the first and second nearest neighbor at layer 10 of the LLAMA model for a single prompt for the full shuffle case and structured case. The dotted vertical lines indicate the average angle between the nearest neighbors in both cases. Right Panel: histogram of the average angle between the first and second nearest neighbor at layer 10 of the LLAMA model in the fully shuffled (orange) and structured case (blue). The histograms are computed from 2244 prompts in each case.

### 4.2.3 NEIGHBORHOOD OVERLAP

We compute the neighborhood overlap at $k_{\mathrm{NN}} = 1$ as a function of layers for the Llama 3 8B model. We choose $k_{\mathrm{NN}} = 1$ because we would like to examine a similar range of scales with ID computed using GRIDE at range scaling = 2. As a consistency check, we also calculate NO for $k_{\mathrm{NN}} = 2$ and $k_{\mathrm{NN}} = 4$ finding similar results (see Appendix B). In Figure 5, we show a random prompt for different levels of shuffling (left panel) and the average over all prompts for the full shuffle and the structured case (right panel). The NO of the shuffled cases is lower than structured case around the layers corresponding to the ID peak, while being statistically similar away from the peak. Again, we can explain this behavior with the help of the generic ratios. On average, if $r_{i,2}/r_{i,1}$ is closer to

unity, which is the case for the shuffled case in correspondence of the ID peak, it implies that the first nearest neighbor is more susceptible to swap with the second nearest neighbor in the next layer, resulting in a smaller neighborhood overlap. Neighborhood overlap has been used as a metric in previous studies working at the prompt-level, see Appendix A for a comparison.

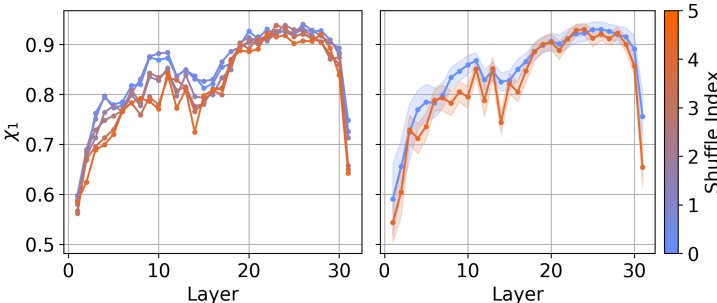

Figure 5: **Neighborhood Overlap.** Left Panel: neighborhood overlap for a single random prompt as a function of model layers for $k_{\mathrm{NN}} = 1$. The colorbar indicates the shuffle index $S$. Right Panel: neighborhood overlap averaged over $2244$ prompts as a function of layers for the full shuffle ($S = 5$) and the structured case ($S = 0$). The shaded regions indicate standard deviation from the mean. All curves have been calculated for the LLAMA model.

### 4.3 COMPARING TOKEN GEOMETRY OF PROMPTS IN DIFFERENT MODELS

In the previous section, we noted that the geometry of internal representations is highly sensitive to shuffled inputs. Having focused on the representations from LLAMA model, we now extend our analysis to include two additional models: MISTRAL and PYTHIA. As described in Section 4.1, we note that PYTHIA was trained entirely on the Pile dataset. Hence, the dataset we consider for experiments, Pile-10K, is a subset of the same dataset on which PYTHIA was trained. While we do not know on which datasets LLAMA and MISTRAL were trained, we can assume that, if present, Pile was not the only dataset used. Therefore, we might expect PYTHIA to have a mildly different signature on our observables compared to MISTRAL and LLAMA. According to what we found in the previous section, we might expect a lower ID peak and a higher NO for PYTHIA.

**Intrinsic Dimension.** We check the ID behaviour for LLAMA, MISTRAL and PYTHIA as a function of layers in Figure 6. On the left panel, we have the ID curve for a random prompt, while on the right panel, we show the mean ID profile across $2244$ prompts. We observe that PYTHIA has a lower ID peak on average than the other two models, though the significance is low.

**Neighborhood Overlap.** Similarly, we calculate NO and show it in Figure 7 as a function of layers for a random prompt (Left Panel) and the average over $2244$ prompts (Right Panel). In this case, we observe that NO is generally higher for PYTHIA with respect to the other two models. The combined behavior of a lower ID peak and a higher NO in PYTHIA is similar to the structured case in the previous section. This might be a consequence of the fact that Pile is more in-distribution for PYTHIA than the other models. However, we note that a more comprehensive analysis would be required to confirm this statement, for instance by performing the analysis on PYTHIA using another dataset.

### 4.4 UNDERSTANDING THE CORRELATION BETWEEN ID AND LOSS

In the previous sections, we have probed the empirical measure across layers through geometric quantities like intrinsic dimension, neighborhood overlap and cosine similarity. This analysis aimed to gather insight on model behaviour, specifically on the relationship of our observations with next token prediction. In this section, we argue that we should expect a positive correlation between the ID of internal representations (a *physical* quantity defined on the token representations) and loss (an information-theoretic quantity defined on the next token prediction probabilities). This correlation is built through 3 steps: first of all, we expect the ID of the final layer, which probes the output measure in the framework of Geshkovski et al. (2024b), to be strongly correlated to the ID of the

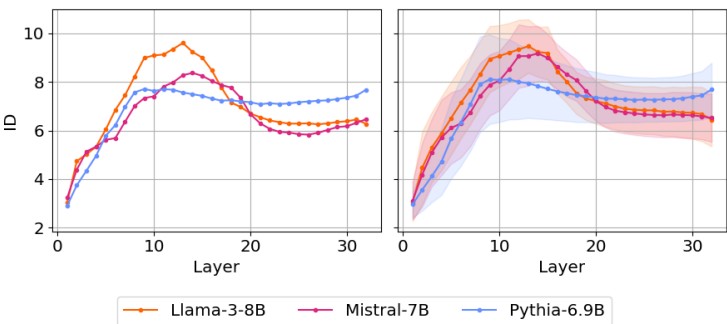

Figure 6: **Intrinsic Dimension.** Left Panel: intrinsic dimension for a single random prompt as a function of layers. Right Panel: intrinsic dimension averaged over 2244 prompts as a function of layers. The shaded regions indicate standard deviation from the mean. The three curves correspond to LLAMA (orange), MISTRAL (magenta) and PYTHIA (blue).

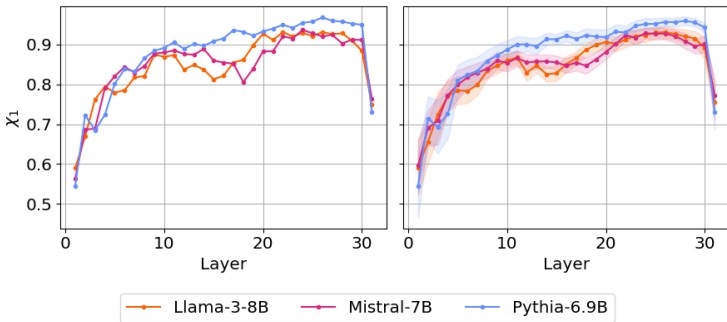

Figure 7: **Neighborhood Overlap.** Left Panel: neighborhood overlap for a single random prompt as a function of layers. Right Panel: intrinsic dimension averaged over 2244 prompts as a function of layers. The shaded regions indicate standard deviation from the mean. The three curves correspond to LLAMA (orange), MISTRAL (magenta) and PYTHIA (blue).

logits, since the unembedding is a linear transformation. [7] Secondly, we expect that the logits ID should be correlated to the average entropy of the next token probabilities $S(X)$, which we refer to as the softmax entropy for brevity. Since this is a non-trivial connection, we elaborate on this in detail in Appendix D. Thirdly, when we consider a large number of tokens as in our case, we can expect the softmax entropy to be almost equal to the cross-entropy loss as seen in Figure 17,

$$\text{loss}(X) = -\frac{1}{N} \sum_i^N \log p_\theta \left( x_i \mid x_{<i} \right) \tag{5}$$

To quantitatively verify the ID-loss correlation, we use the Pearson correlation coefficient ($\rho$), which is defined as the ratio between the covariance of two variables and the product of their standard deviations. We compute the Pearson correlation coefficient between the ID and the model's loss across layers for the population of 2244 prompts and show the result in Figure 8. We find a high correlation in all three models, particularly around the ID peak. The connection between the loss (log perplexity) and ID was discussed in (Cheng et al., 2023) where the correlation was calculated between the maximum ID of the dataset of the last token representations and the log of dataset perplexity in Fig. 2 of (Cheng et al., 2023). We get a correlation in a similar spirit, however at the finer level since it reveals a correlation at the prompt level, see more details in Appendix A.

---

[7]We have verified this statement quantitatively, see Appendix D. Even though we sketch an argument as to why we expect a positive correlation between ID and loss, we don't yet comment on why ID around the peak is more correlated to the loss than the output ID. We leave this for future work.

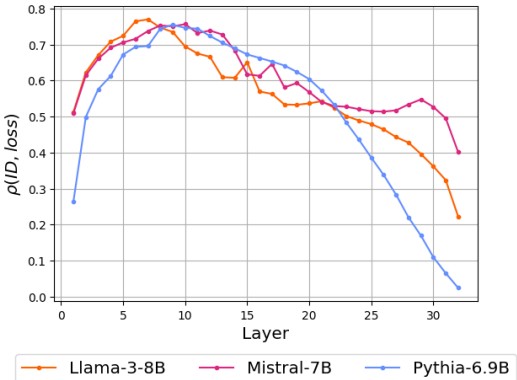

Figure 8: **Correlation between intrinsic dimension and loss.** Pearson coefficient between intrinsic dimension and model loss for different models as a function of layers. The shaded regions indicate standard deviation from the mean. The three curves correspond to LLAMA (orange), MISTRAL (magenta) and PYTHIA (blue). The $p$-values for the Pearson coefficients in this plot are below $0.01$ except the last layer in PYTHIA.

## 5    CONCLUSIONS

The primary aim of this study was to connect different approaches to the interpretability of LLMs. Our strategy towards this goal was to examine the geometric structure of token-level representations across the layers of these models and to connect it to the probability distribution of the next token prediction. We employed three key metrics: cosine similarity, intrinsic dimension, and neighborhood overlap, to capture different aspects of this geometric structure. Our findings revealed that the intrinsic dimension of token representations peaks in the early to middle layers, with higher peaks in shuffled data, i.e. when syntactic and semantic structures are disrupted. Additionally, cosine similarity among tokens increases with shuffling, suggesting greater alignment of token vectors. The neighborhood overlap metric showed that structured data maintains more coherent token neighborhoods across layers, while increased shuffling reduces this consistency, reflecting the model's sensitivity to the input structure. We observe these features consistently across different models. All these analyses converge into the key finding of this paper, which is the correlation of the ID of token representations to the model's loss, implying that ID could be an important metric for evaluating model performance across different models.

This correlation should be notably significant during the training process. As demonstrated at the prompt level in previous research Cheng et al. (2024), and confirmed by our findings at the token level, see Appendix E, ID remains largely constant and low in the early training stages and is not correlated with loss, but it increases as training progresses. At the token level, we observe that the ID tends to rise due to enhanced model expressivity, while there is also a tendency for ID to decline as the minimization of loss improves. Indeed, as seen in Figure 19, ID initially rises and then shows a slight decrease after checkpoint 64K. We believe it would be intriguing to explore these aspects in greater depth, but we defer this investigation to future work.

Experiments could be improved in several directions: first, we computed our observables at low ranges of nearest neighbors. For a more holistic approach, a multiscale analysis can reveal further relations among these observables. Secondly, the differences in distribution patterns for structured versus shuffled data, as suggested by cosine similarity and ID studies, might encode essential information on how tokens are distributed in space in the two cases. It is interesting to consider other geometric observables and understand their relation to the next token probabilities. These targeted explorations could provide practical applications for the design and training of LLMs, potentially leading to more interpretable and efficient models. While we show that the geometry of tokens encodes the next token prediction loss, we also potentially provide an unsupervised tool to interpret how the model processes a given prompt. An interesting avenue in this regard can be a more in-depth analysis of the lower ID peak in PYTHIA which we reserve for future work.

## 6 REPRODUCIBILITY

The experiments were run on an NVIDIA H100 GPU with 94 GB memory. All the results contained in this work are reproducible by means of an anonymized repository that can be found at https://anonymous.4open.science/r/token_geometry-DB87.

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

## A   HOW IS TOKEN GEOMETRY DIFFERENT FROM PROMPT GEOMETRY?

Previous work Ansuini et al. (2019); Doimo et al. (2020a); Pope et al. (2021); Valeriani et al. (2023); Cheng et al. (2023; 2024); Cheng & Antonello (2024) have studied internal representations from a geometric point of view by considering point clouds of last token representations as observable. While the approach is similar in spirit, token-level and prompt-level measures of intrinsic dimension probe different manifolds and thus different features of LLMs.

While prompt-level and token-level ID profiles exhibit similar behavior qualitatively, e.g. they peak in early-middle layers, there is a notable difference in the shuffled and unshuffled prompts. At the prompt level, we see that the unshuffled ID has a more prominent peak than the shuffled ID, whereas it is the other way round at the token level. This difference between token and prompt level ID curves offers a window to understand the difference between the token and prompt geometries. The core reason for this diverging behavior is that we are looking at different manifolds and thus observing two distinct behaviors: token level ID is correlated to the input perplexity (measured using the cross-entropy loss) and the prompt level ID is a measure of the semantic information Valeriani

et al. (2023); Cheng et al. (2024). Given a dataset of the prompts with a high perplexity at the token level such as in the shuffled case, we can expect the last token representations are less likely to share semantic content, leading to a lower intrinsic dimension at the prompt level. At the token level, the lesser prominence of the peak of the unshuffled case can be explained using the ID loss correlation. Since the loss is expected to be lower for the unshuffled prompts, we can expect their ID peak to be less prominent than that of the shuffled prompts. In the following section, we examine the prompt-level and token-level geometries for both shuffled and unshuffled cases at the peak layer, where the differences are most pronounced.

**Token and prompt geometries of shuffled and unshuffled prompts at layer 11**

For the prompt-level analysis, we use a corpus of $2242$ prompts (the same corpus used for the token level analysis), drawn from Pile-10K and consisting of prompts with at least $1024$ tokens. The last token representations are extracted from these prompts as follows - we choose tokens at positions $512$ through $532$ that result in a 20-token sequence for the unshuffled case[8]. We randomly permute aforementioned the 20-token sequences in the shuffled case and obtain the last token representations. The token-level analysis is done on prompt number 3218 from the Pile-10K dataset.

In Figure 9, we plot the tSNE projections of the shuffled and unshuffled along with ID for different scalings at both the prompt and token levels. We notice that in both levels, the shuffled and unshuffled representations lie on separate manifolds Sarfati et al. (2024).

- Prompt level - We can't learn much from the tSNE projections at the prompt level since the data is very high dimensional $\mathcal{O}(30)$. However, we can see a clear difference in the ID profiles - the unshuffled case has a higher ID from scaling $= 16$ onwards.

- Token level - In Figure 9b, we see that the unshuffled prompts form a more continuous manifold than the shuffled prompts, i.e. there are more "gaps" in the shuffled case. In the unshuffled case, we can also see that the tokens tend to cluster by their position in the sequence[9], i.e. if two tokens are nearby in a prompt, they are also close in the embedding space (perhaps due to positional embedding). This happens in the shuffled case as well but to a lesser degree. The intrinsic dimension for the unshuffled case is smaller at low scales (until scaling $= 8$) and higher at large scales (scaling $= 128$ onwards) than the shuffled one.

**Token level ID is more strongly correlated to surprisal** Since there is an extensive amount of work done for the case of Opt-6.7B at the prompt level, we compare the token level results to the prompt level for Opt-6.7B. Before proceeding here is a summary of the prompt level results from Cheng et al. (2023) and Cheng et al. (2024) that are relevant for our comparison.

- In Cheng et al. (2023), the authors show a positive Spearman correlation of $0.51$ for Opt-6.7B (Figure 2a in Cheng et al. (2023)) using the ID estimator Expected Simplex Skewness (ESS) Johnsson et al. (2015).

- Using TwoNN, they do not get a statistically significant correlation (Figure E5 in Cheng et al. (2023)). This is expected if we extrapolate from the GRIDE scale analysis of Figure 9a since the shuffled and unshuffled prompts were not distinguishable at low range scalings at the prompt level.

- An analysis at a higher range scaling is done in Cheng et al. (2024) where they show a **negative correlation** with surprisal (Figure 6 in Cheng et al. (2024)) with a relatively less statistical significance since it has a high $p$-value $= 0.09$.

On the other hand, using the token-level approach, we measure a **layerwise positive correlation** with surprisal[10]. We summarize the results in Table 1.

---

[8]This is a simplified setup of the experiments in Cheng et al. (2024).

[9]Note that the last tokens do not cluster according to the prompt index at the prompt level and we do not expect it to happen. This gives a rough idea of how representations should look in case they are not related to the sequence position.

[10]Surprisal, cross-entropy loss, log-perplexity are used interchangeably in this context since we are mainly providing a qualitative picture in this section.

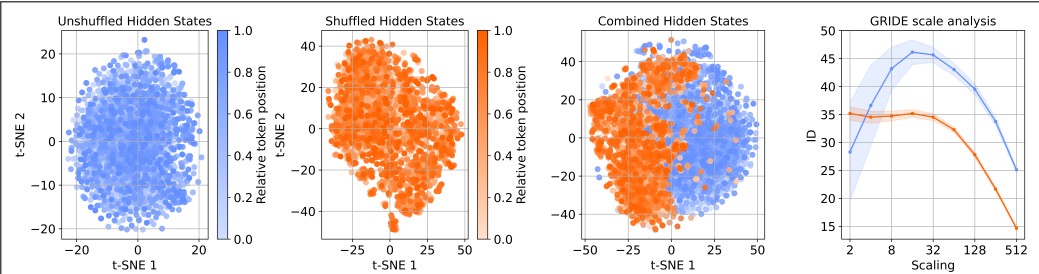

(a) **Prompt level.** Left and middle left panels: The tSNE plots are of the 2242 last token representations in both the unshuffled and the shuffled cases. Middle right panel: The combined tSNE is made of 4484 last token representations from both the shuffled and unshuffled cases. Right panel: ID for different scalings calculated by randomly sampling 1024 last tokens from the 2242 tokens averaged over 50 samplings. The shaded area is the standard deviation around the mean.

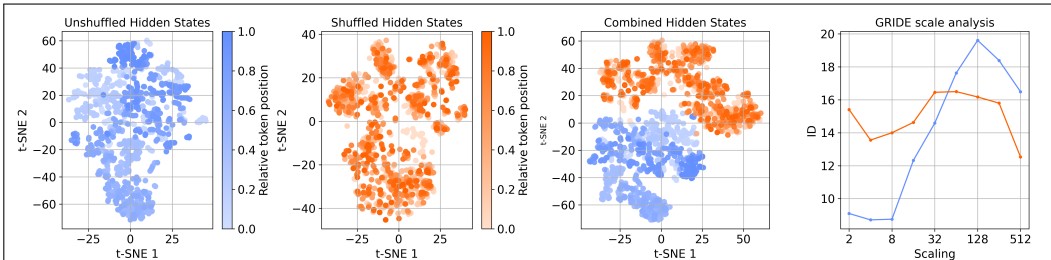

(b) **Token level.** Left and middle left panels: The tSNE projections for the token representations for the unshuffled and shuffled case of prompt number 3218 from Pile-10K. Middle right panel: Token representations of both the unshuffled and shuffled case made of 2048 tokens. Right panel: the ID for different scalings is calculated using 1024 tokens for both cases.

Figure 9: **Prompt geometry and token geometry.** Comparing prompt level (top panel) and token level (bottom panel) geometry at layer 11. All the plots are obtained using the representations from LLAMA.

|  | Prompt level (ESS) | Prompt level (2NN) | Prompt level (high scaling) (many models × corpus) | Token level (2NN) | Token level (scaling = 8) |
|---|---|---|---|---|---|
| Spearman $\rho$ | 0.51 | 0.13 | -0.46 | 0.69 | 0.73 |
| $p$-value | 0.01 | 0.5 | 0.09 | < 0.01 | < 0.01 |

Table 1: Summary of Spearman correlations between ID and loss from prompt and token level analysis for Opt-6.7B. The results for token level are from Figure 10 and the prompt level are from Cheng et al. (2023) and Cheng et al. (2024).

## B    CONSISTENCY CHECKS FOR THE SHUFFLE EXPERIMENT

In this section, we show the consistency of the results that were discussed in the Section 4.2.

**Intrinsic Dimension.** For the case of intrinsic dimension, we show the ID profiles of 6 random prompts sampled. It can be seen from Fig. 11 that the shuffled ID (orange) peak is always higher than the structured ID peak (blue) even though the degree of difference varies across prompts. We also verify that this behavior is consistent across models in Fig. 12.

**Neighborhood Overlap.** We compute NO for the average over all prompts of the full shuffle and the structured case using $k_{\text{NN}} = 2$ (left panel) and $k_{\text{NN}} = 4$. Results are consistent with what discussed in 4.2.3.

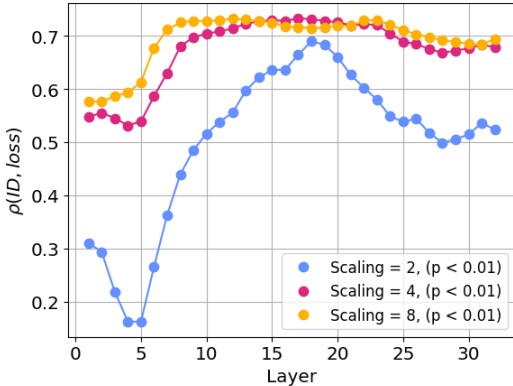

Figure 10: Spearman correlation between ID and loss for Opt-6.7B for different range scalings at the token level as a function of layers.

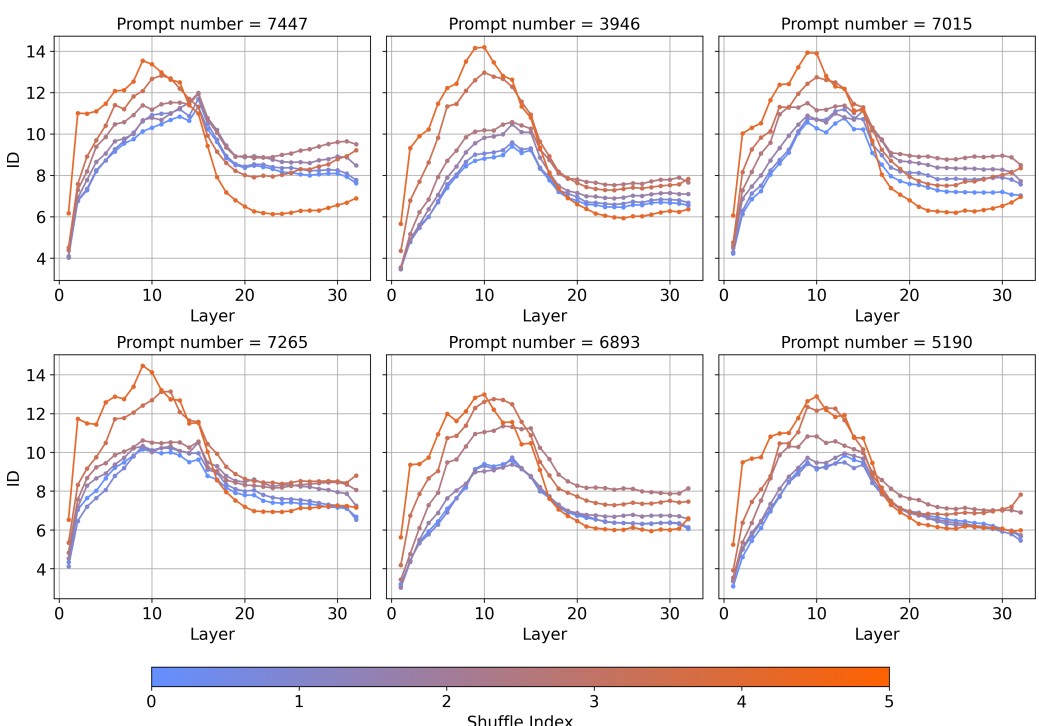

Figure 11: Intrinsic dimension profiles of 6 random prompts for LLAMA. The prompts are taken from the filtered version of Pile described in the dataset section 4.1, where the prompt numbers refer to the Pile-10K dataset.

## C    SCALE ANALYSIS FOR GRIDE

In this section, we analyze the different choices of range scaling for the GRIDE algorithm discussed in Section. 3. The prompts we analyze have $N = 1024$ tokens and in Fig. 14, we check the dependence of ID estimate on range scaling $\in \{2, 4, 8, ..512\}$ for a single prompt on different models. This is to illustrate the scale dependence of a single prompt that we consider throughout the text.

In the main text, we focus on range scaling $= 2$ and here we extend the analysis to range scaling $= 4$ and 8. In Figure 15a, we find that PYTHIA's peak is more comparable to LLAMA and MISTRAL as the range scaling increases. In Figure 8, we notice that the correlation to loss becomes stronger for range scaling $= 4$ and 8.

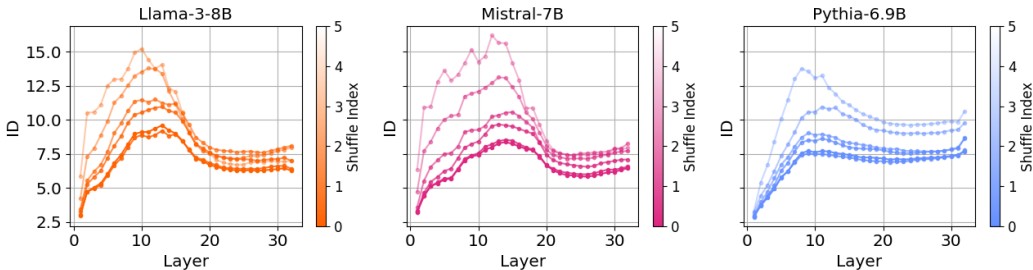

Figure 12: The ID profiles for prompt number 3218 from Pile-10K for different models. Lighter colors represent a higher shuffle index and a darker color is closer to the structured prompt.

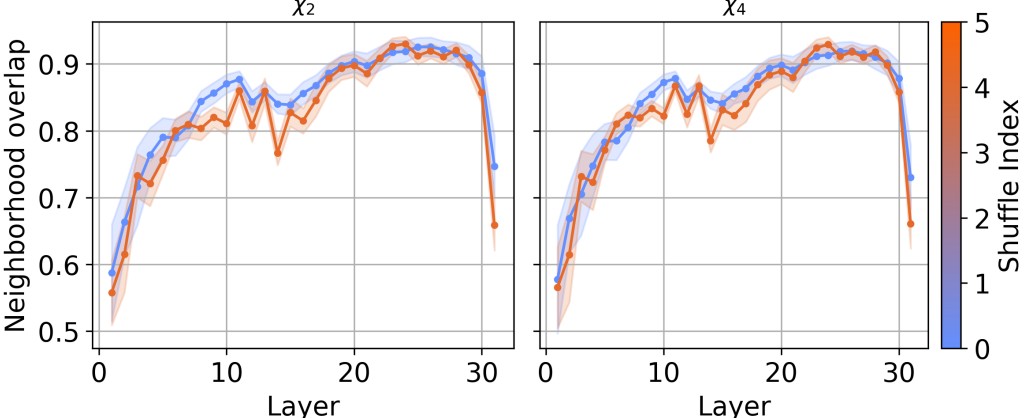

Figure 13: Neighborhood overlap at $k_{\mathrm{NN}} = 2$ (left panel) and $k_{\mathrm{NN}} = 4$ (right panel) as a function of layer for the full shuffle and structured case for the average over all prompts for LLAMA.

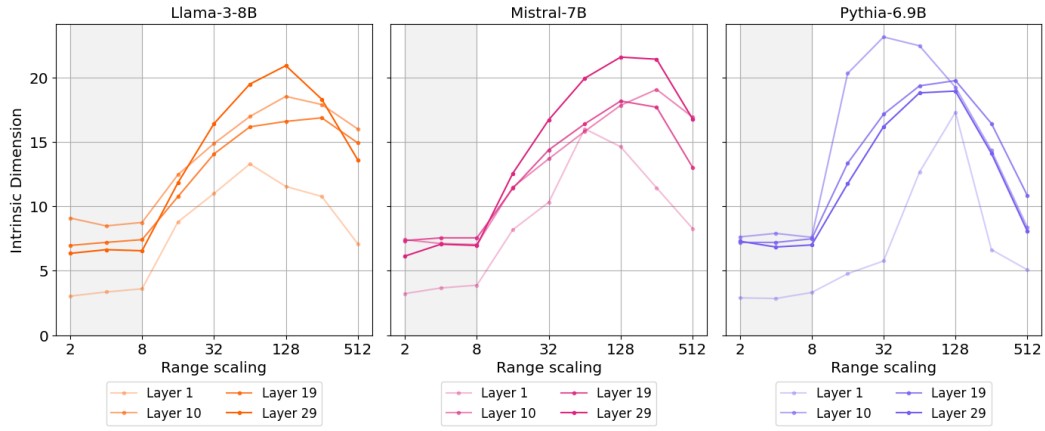

Figure 14: Scale analysis for GRIDE estimation across models for a single prompt (prompt number 3218) for different layers. The early layers are given by lighter colors and the late layers are given by darker colors.

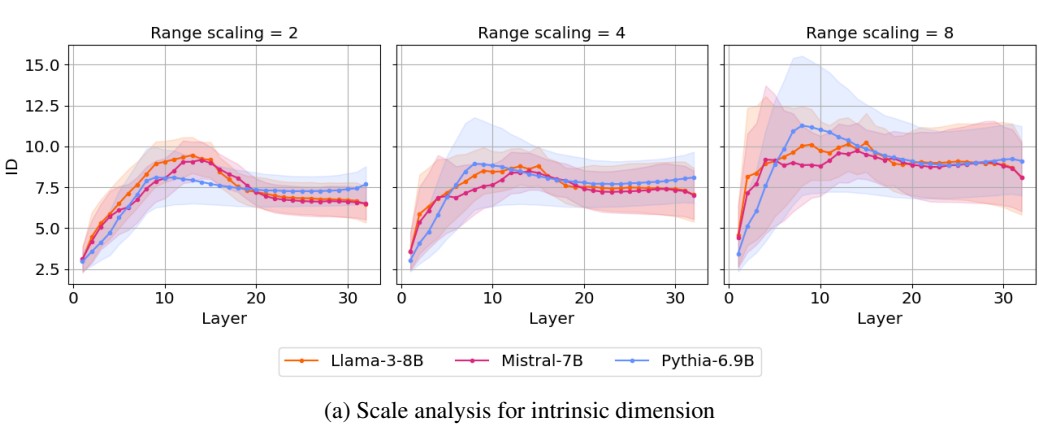

(a) Scale analysis for intrinsic dimension

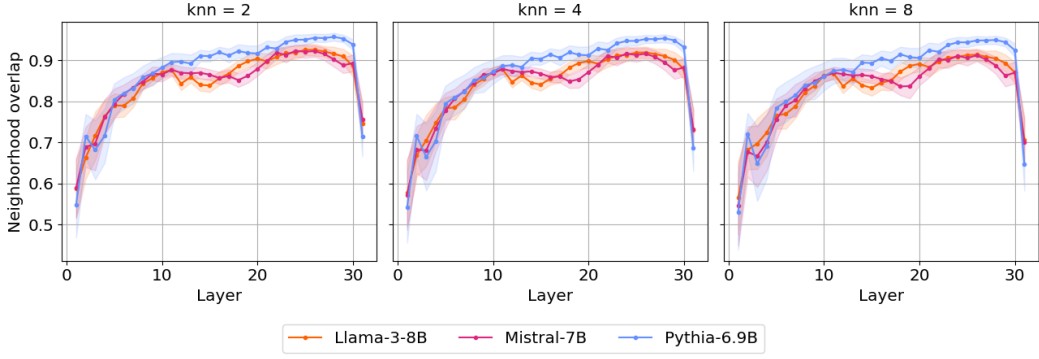

(b) Scale analysis for neighborhood overlap

Figure 15: **Scale analysis for intrinsic dimension and neighborhood overlap.** Top Panel: The ID profile averaged over 2244 prompts for range scaling $= 2, 4, 8$, with shaded regions indicating the standard deviation from the mean. Bottom Panel: The neighborhood overlap profile averaged over 2244 prompts for range scaling $= 2, 4, 8$, with shaded regions indicating the standard deviation from the mean.

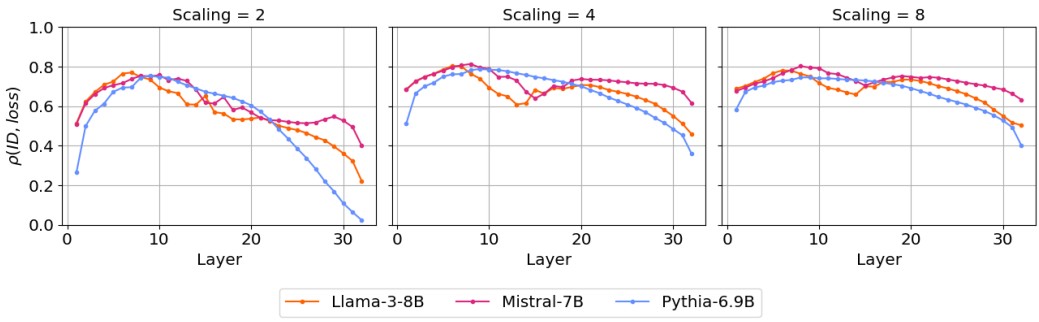

Figure 16: **Correlation between intrinsic dimension and loss at different range scalings.** Pearson coefficient between intrinsic dimension and model loss for range scalings $= 2, 4$ and $8$ for different models.

## D    MORE DETAILS ON THE CORRELATION BETWEEN ID AND LOSS

In section 4.4, we have discussed 3 steps to correlate the ID of tokens in internal representations to the model's loss. We empirically verify these steps in Figure 17.

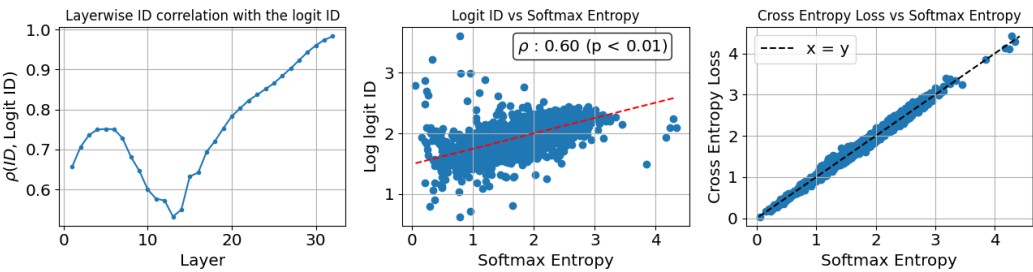

Figure 17: **Correlating Intrinsic Dimension at the hidden layers to Cross Entropy Loss** The points in the following plots are calculated using the $2244$ prompts considered in this paper for the LLAMA model. We use scaling $= 4$ to compute the ID for the logits and the hidden layers. We understand the correlation in 3 parts - (a) Left Panel: finding the correlation between the logit ID and ID at the hidden layers, (b) Middle Panel: correlating the logit ID to the softmax entropy and (c) Right Panel: comparing the softmax entropy to the cross entropy loss.

### D.1    ANALYTICALLY RELATING THE INTRINSIC DIMENSION AND THE SOFTMAX ENTROPY

Given the logits $\mathbf{X} = (x_1, x_2, ... x_D)$ as the input to a softmax layer, we get the associated probability distribution given by

$$p(\mathbf{X})_i = \frac{e^{x_i}}{\sum_{j=1}^{D} e^{x_j}} \tag{6}$$

where $p(\mathbf{X})_i$ is the probability of the $i^{\text{th}}$ *word* (in the $D$-dimensional vocabulary) to be the next token. We can now write the associated entropy $S(\mathbf{X})$ of this probability distribution -

$$S(\mathbf{X}) = - \sum_{i=1}^{D} p(\mathbf{X})_i \log p(\mathbf{X})_i = \left( \log \sum_{j=1}^{D} e^{x_j} - \frac{\sum_{j=1}^{D} x_j e^{x_j}}{\sum_{j=1}^{D} e^{x_j}} \right) \tag{7}$$

Now we want to ask the following question - *If $\mathbf{X}$ is sampled from a $\mathcal{D}$-dimensional manifold $\mathcal{M}$, what is the expected value of the entropy $S(\mathbf{X})$?* In the upcoming paragraph, we look at a simpler case when the manifold we consider is the unit box, i.e. $\mathcal{M} = [0, 1]^D$.

**Evaluating softmax entropy as a function of dimension for the unit box** Here we take a simplified example - the unit box and understand the trend of expected softmax entropy with respect to the

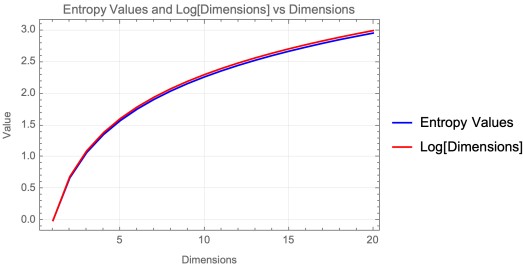

Figure 18: Comparing expected entropy $\langle S \rangle$ for the unit box to $\log D$ until $D = 20$

dimension of the box -

$$\langle S \rangle_{[0,1]^D} = \int_{[0,1]^D} d\mathbf{X} \, S(\mathbf{X}) = \int_0^1 \cdots \int_0^1 d\mathbf{X} \left( \log \sum_{j=1}^D e^{x_j} - \frac{\sum_{j=1}^D x_j e^{x_j}}{\sum_{j=1}^D e^{x_j}} \right) \quad (8)$$

where $\mathbf{X} = (x_1, x_2, ... x_D)$. We numerically evaluate this integral using NIntegrate on Mathematica and compare the result with $\log D$ in Figure (18) and notice that $\log D$ is a good approximation to the above integral. Hence we have related an information-theoretic quantity the expected softmax entropy ($\langle S \rangle_{[0,1]^D}$) to a physical quantity $D$ - the dimension of the unit box, where we have

$$\langle S \rangle_{[0,1]^D} \sim \log D \quad (9)$$

## E  TOKEN-LEVEL ID DURING TRAINING

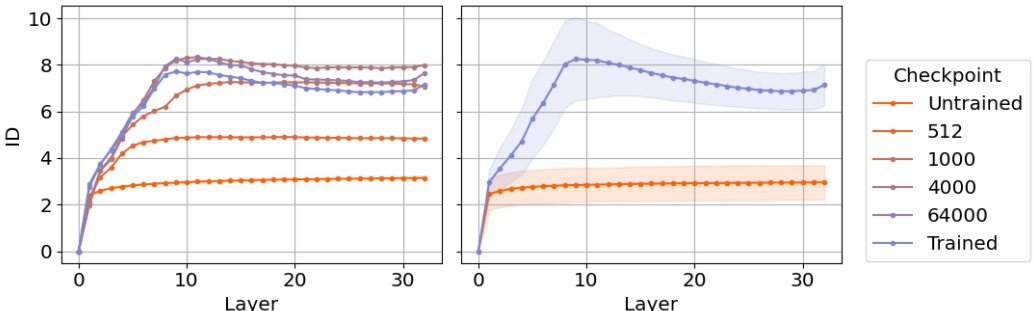

Figure 19: **Intrinsic Dimension profile over training for PYTHIA.** Left Panel: intrinsic dimension profile for a single random prompt as a function of layers for different levels of training. Right Panel: intrinsic dimension averaged over 50 prompts as a function of layers for the untrained (orange) and trained (blue) model. The shaded regions indicate the standard deviation from the mean.

