# OpenReview forum: "The Geometry of Tokens in Internal Representations of Large Language Models"
_ICLR.cc/2025/Conference — Submitted to ICLR 2025_

### Official Review · Reviewer_1NWE · 2024-11-02

**Soundness:** 2
**Presentation:** 2
**Contribution:** 2
**Rating:** 5
**Confidence:** 3

**Summary:**

This study investigates the geometric structure of token-level representations across layers in large language models using three metrics: cosine similarity, intrinsic dimension, and neighborhood overlap. Unlike existing studies that typically summarise an entire prompt as a single point in space, this work measures geometry at the token level, offering a more granular perspective. To compare different structures, measurements are performed on both an "in-distribution" structured case and an "out-of-distribution" shuffled case, providing insights into how representation geometry varies across distributions.

**Strengths:**

1. Finer-Scale Analysis of Internal Representation Geometry: This study advances prior research by analyzing the geometric structure of internal representations at the token level rather than treating prompts as a single point in the representational space.

2. Token Shuffling as a Comparison Baseline: The study introduces a shuffled-token case to represent out-of-distribution data, offering a valuable contrast to structured, in-distribution input. This approach enables the revealing of changes in token representation when standard sequence patterns are disrupted. It provides a clearer view of how language models process structured versus unstructured input.

**Weaknesses:**

1. Metrics Similarity and Limited Novelty in Findings: The study largely employs metrics similar to those in prior research, such as cosine similarity and intrinsic dimension. Consequently, while the paper provides an interesting token-level perspective, many findings echo previous studies without introducing substantially new insights.

2. Unclear Utility for Model Interpretability: While the paper explores token-level geometry, its implications for model interpretability remain uncertain. A connection between the identified geometric patterns and practical insights into model behaviors would be appreciated.

3. Questionable Plausibility of Some Claims: Certain claims, like the difference in Neighborhood Overlap between structured and shuffled cases, appear unsupported by the presented data. For instance, Figure 4 does not show a clear difference in NO between the two cases, particularly in the average result shown in the right panel. Instead of concluding that the shuffled case consistently has a lower NO, it would be valuable to investigate this observed similarity further to determine if it reflects a limitation in the metric’s sensitivity or a more nuanced interaction in token representations between the cases.

**Questions:**

N/A.

---

> ### Author Response · Authors · 2024-11-22
> **Specialized Response to Reviewer 1NWE**
>
> ## Reviewer 1NWE
>
> >Metrics Similarity and Limited Novelty in Findings: The study largely employs metrics similar to those in
> prior research, such as cosine similarity and intrinsic dimension. Consequently, while the paper provides
> an interesting token-level perspective, many findings echo previous studies without introducing substantially
> new insights.
>
> We believe we have answered this feedback in our general response. In short, while we do use
> previously employed metrics to probe the geometry of internal representations, we use them with a specific
> goal in mind, which is probing the empirical measure of internal representations, as a way to connect to the
> probability distribution of the next token. The token-level perspective provides different insights than previous
> applications of these metrics, as listed in general response and expanded on Appendix A.
>
> >Unclear Utility for Model Interpretability: While the paper explores token-level geometry, its implications for
> model interpretability remain uncertain. A connection between the identified geometric patterns and practical
> insights into model behaviors would be appreciated.
>
> The main implication for model interpretability is the correlation of ID with loss, which we have made
> stronger now by connecting it more explicitly to the probability distribution of next token.
>
> >Questionable Plausibility of Some Claims: Certain claims, like the difference in Neighborhood Overlap between structured and shuffled cases, appear unsupported by the presented data. For instance, Figure 4 does
> not show a clear difference in NO between the two cases, particularly in the average result shown in the right
> panel. Instead of concluding that the shuffled case consistently has a lower NO, it would be valuable to investigate this observed similarity further to determine if it reflects a limitation in the metric’s sensitivity or a
> more nuanced interaction in token representations between the cases.
>
> In the text, we discuss the difference between NO for structured and shuffled cases *around layers
> where ID peaks*, which is were the difference is significant, as shown by the non-overlapping standard deviations of the two curves. Here we should also notice that we rerun the experiment including all 2244 in
> the estimation of the standard deviation. We do agree that the difference is not seen across all layers. This
> might be due to, as the referee proposes, a limitation in the metric’s sensitivity to shuffling outside the ID peak
> layers.

---

### Official Review · Reviewer_2YbW · 2024-11-02

**Soundness:** 3
**Presentation:** 3
**Contribution:** 3
**Rating:** 6
**Confidence:** 4

**Summary:**

The paper describes observational data about the geometry of token representations in language models, focusing on attributes related to the dimensionality and layer similarity. The authors relate these variables to information on loss and presence of test data in the training set.

**Strengths:**

Geometric analysis of transformers holds great promise for interpretability, so this type of contribution is important. In addition, it's terrific to see careful observational data. The results themselves are interesting, but please see the next section for important caveats about novelty.

The technical exposition is fairly clear, and I found the review of various geometric measures useful. The four main results fit together, telling an interesting story that suggests that the level of information reduction is key to transformer performance. We see this both in the fact that shuffled inputs seem to have higher ID, and that ID shows a correlation with model loss.

The neighborhood consistency metric is also interesting, and provides more data for theorists looking to explain how different layers interact. The comparison with shuffling is useful to provide a baseline, indicating that the consistence results relate to "real" processing, rather than being an artifact of the architecture.

Overall, I certainly appreciate this type of observational result. I don't know that we can draw any strong conclusions from these experiments, but that shouldn't be necessary for acceptance: this will be helpful to future researchers as they build out theories of how transformers work.

**Weaknesses:**

My main confusion with this paper is understanding how much of an advance it is over the Cheng et al 2023 paper it references: https://arxiv.org/abs/2405.15471
It feels like the methodology and goals are extremely similar. Both papers study geometric aspects of internal token representations, with at least three target models in common. Both make comparisons between regular vs. shuffled prompts, look at the relationship between loss and intrinsic dimension, and compare ID and representation similarity between layers (although with slightly different methods). The results seem closely related. As I read it, the current paper feels like a relatively small advance over the Cheng et al paper, but I'm happy to be corrected.

Separately, I'm not convinced the "shuffled" condition is the best (or at least, only) baseline. I think it would be interesting to compare geometry with an randomly initialized network. (If this is in the literature already, I'd recommend noting this fact in the related work section.)

Another confound I wondered about: the results that prompts with higher ID have higher loss seem ambiguous. Could it be possible that high-ID prompts are just more intrinsically complicated in some way—maybe even in a trivial way, such as length? If so, it's not clear whether we can conclude anything useful from ID.

Writing: I think the authors could simply delete the first paragraph of the intro: it's not really necessary. In Figure 3, the left panel is mislabeled as "Angle".

UPDATE: I have bumped my ratings up 1 point to reflect the authors' changes, especially the clarifications of the contributions,

**Questions:**

I'd like to see a systematic review of how the results here compare to Cheng et al. This may help make clear which parts of the current paper represent a new advance.

---

> ### Author Response · Authors · 2024-11-22
> **Specialized Response to Reviewer 2YbW**
>
> ## Reviewer 2YbW
>
> > My main confusion with this paper is understanding how much of an advance it is over the Cheng et al 2023
> paper it references: https://arxiv.org/abs/2405.15471 It feels like the methodology and goals are extremely
> similar. Both papers study geometric aspects of internal token representations, with at least three target
> models in common. Both make comparisons between regular vs. shuffled prompts, look at the relationship
> between loss and intrinsic dimension, and compare ID and representation similarity between layers (although
> with slightly different methods). The results seem closely related. As I read it, the current paper feels like a
> relatively small advance over the Cheng et al paper, but I’m happy to be corrected.
>
> We thank the reviewer for this feedback, indeed a careful comparison to the mentioned paper helps
> improving our work. We have included this comparison in Appendix A.
>
> >Separately, I’m not convinced the ”shuffled” condition is the best (or at least, only) baseline. I think it would
> be interesting to compare geometry with an randomly initialized network. (If this is in the literature already,
> I’d recommend noting this fact in the related work section).
>
> We added an experiment on Pythia at different checkpoints of learning, including the untrained case,
> see Appendix E.
>
> >Another confound I wondered about: the results that prompts with higher ID have higher loss seem ambiguous. Could it be possible that high-ID prompts are just more intrinsically complicated in some way—maybe
> even in a trivial way, such as length? If so, it’s not clear whether we can conclude anything useful from ID.
>
> We look at the ID of sequences of fixed length = 1024, hence the ID values are not impacted by the
> sequence length in our experiments. Moreover, in the shuffle experiment, we fix both the sequence length and
> the tokens themselves and we find a sensitivity of ID by changing the ordering of tokens which suggests that
> prompts with the same length and unigram frequency of tokens can have different IDs.
>
> >I’d like to see a systematic review of how the results here compare to Cheng et al. This may help make clear
> which parts of the current paper represent a new advance.
>
> We have briefly introduced a comparison in the general response. For a more thorough analysis, we
> have added a dedicated appendix, see Appendix A.

---

### Official Review · Reviewer_bby9 · 2024-11-04

**Soundness:** 4
**Presentation:** 2
**Contribution:** 2
**Rating:** 8
**Confidence:** 3

**Summary:**

The paper considers several questions relating to the geometry of LLM embedding spaces across layers, using intrinsic dimensionality (ID), neighborhood overlap (NO), and cosine similarity to study and characterize LLM embedding geometry. Extensive experiments are performed across three leading open-source LLMs and variable levels of noise applied to inputs (by shuffling), yielding several key results: ID usually peaks in early to middle LLM layers; inputs that are out-of-distribution (OOD) with respect to LLMs show higher ID peaks and lower NO; and average cosine similarity between tokens in each layer increases with noise (i.e., with a higher shuffling index).

**Strengths:**

The paper is generally quite well-written, and performs extensive and thorough experiments across several geometric measures, LLMs, and levels of noise (shuffling) applied to inputs. All experiments are methodologically sound, and most are original and offer novel results. While there are a number of key questions and potential weaknesses of the paper (as discussed below) -- most importantly, that the motivation and contribution are not clear -- it seems plausible that the approach and findings of this work could be significant if these issues are resolved.

**Weaknesses:**

The primary weakness is that **the motivation and contribution of the presented analysis is not clear.**
- At a high-level, the general motivation for improving our understanding of LLMs *by studying the geometry of their embedding spaces* is not clear. That is, how would our broader understanding of LLMs or other foundation models be advanced by studying the geometric properties considered in this work? For example, at a foundational level, are there questions about how or what any given LLM has learned from its training data that could be answered on this basis, or could predictions be made about likely behaviors (outputs)? Or at an applied level, are there elements of LLM training or deployment that could be improved given the findings of this work?
- At a lower-level, the specific importance of the studied properties (ID, NO, and cosine similarity) is not clear, nor is it clear why LLMs are studied at different levels of noise (shuffling) applied to inputs. There are also other more detailed questions about experiments that need to be clarified, as discussed in more detail below.
- (Note: while some brief thoughts regarding the motivation are laid out in the last paragraph of the conclusion, they are incomplete; and this also needs to be made clear early on in the paper (i.e., in the introduction), as otherwise, on a single forward pass of the paper, it's impossible to interpret or contextualize the results throughout the paper.)

Additionally, **most of the experiments and results are not contextualized with respect to prior work** in this area. That is, with the exception of two experiments -- cosine similarity (sec 4.2.1) and "correlation between ID and loss" (end of sec 4.3) -- it is not clear how the design and findings of each experiment relate to those of prior work, hindering the reader's understanding of the contribution offered by this work and its potential impact on the broader research literature.

Finally, **most of the broader literature studying the internal representations learned by LLMs is not discussed as related work.** Only a narrow sliver of work studying LLM geometry is mentioned -- i.e., work studying embeddings as particle systems, or according to ID, NO, or cosine similarity -- but there are other broad and longstanding bodies of work (such as probing, circuit discovery, feature attribution, etc.) that also aim to mitigate the "black box problem" and understand the internals of foundation models such as LLMs. (For a few relevant surveys of such work, see, e.g., https://arxiv.org/abs/2408.05859 or https://arxiv.org/abs/2404.14082 -- I will refer to this broad category of work studying the black box problem of foundation model internals as "mechanistic interpretability", or MI for short.) While it is not necessary to summarize all such work in detail, and entirely appropriate to focus on the most relevant work studying embedding geometry, it is nonetheless important to position this work in the broader literature working toward understanding LLM representations -- otherwise, as noted above, the contribution and broader impact of this work is not clear.
- I suggest that, at minimum, a brief overview of (1) what is studied in MI, and (2) the relationship between MI and the geometric questions studied here, should be discussed.
    - In the ideal case, it would also be useful to discuss what insights could be provided by the kind of geometric analysis considered in this work that are *complementary* to MI, that could *validate or contradict* key findings in MI, etc.
    - E.g., there is currently a hot debate in MI over "how linear" LLM representations are -- see, e.g., https://arxiv.org/abs/2406.01506 and https://arxiv.org/abs/2405.14860. While it is of course not the goal of this paper to address such questions, I am sharing this context to provide a typical example of how questions around embedding geometry are discussed in MI, which might be useful for the authors in contextualizing their work.

I also noticed a few minor errata:
- line 183: typo/missing words
- sec 4.2.3: need to mention which figure is being discussed -- it is all in reference to Figure 4, correct?
- line 487-489: incomplete sentence

Finally, I would like to note that -- given that the experimental work carried out in this paper is quite good, and these weaknesses are primarily a matter of framing, clarity of contribution, providing adequate context, etc. -- I am quite open to raising my rating if the authors are able to address the above weaknesses and answer the questions provided below.

**Questions:**

In addition to the most important questions inherent in the weaknesses highlighted above (e.g., what is the motivation, relationship to experiments/findings of the most closely related work, relationship with MI more broadly, etc.), there are a number of other questions that need to be answered, which I will list by category below.

Conceptual questions:
- Why is it meaningful to "progressively disrupt the syntactic and semantic structure" of text by increasing the amount of token shuffling? What would we expect to happen as text is shuffled? And why do we care about this, given that there is no reason that one would expect texts to be shuffled before passing them to LLMs?
    - While there is some discussion on this topic in lines 490-494, it does not answer any of these questions -- that is, while naturally one would expect cosine and ID to change when we shuffle the input, why is this an interesting/important finding? Why is "further probing of these distributions and their relation to the model's processing dynamics" essential?
- How do the approaches studying embedding geometry from the perspective of "particles in dynamical system" (Geshkovski et al., 2023; Cowsik et al., 2024) relate to this work?

Experimental design:
- Why are there 50 prompts maximum for all experiments in sec 4.2 (but not 4.3)? This is an exceptionally small-sample experiment in context of LLMs and the Pile dataset being sampled from -- is the number being kept so small because one or more of the experiments is particularly computationally expensive? If so, which ones, and what is the main bottleneck? Do we expect results to look similar for a larger sample?
    - Additionally, how how are the 50 prompts sampled from the 2252 prompts after filtering (per line 190-191)? If the sampling is not purely random, then the sampling procedure must be explained and motivated.
- A major motivator for experiments in sec 4.3 is that the Pile is used to train Pythia but not Mistral or LLAMA; but is it actually public information whether the Pile was used to train LLAMA 3 8B or Mistral? This uncertainty could carry major implications for the findings in sec 4.3. (Per the model papers cited by the authors in sec 4.1, neither model's training data is disclosed; and my understanding is that this information has not been made public since the original model papers, either.)
    - If I am correct and it is not known whether either model is trained on the Pile, then this should be made clear in the paper (both in sec 4.1 and the uncertainty this carries for the results throughout sec 4.3); and otherwise, any source disclosing these models' training data should be cited.

Findings:
- In figure 5, why do we see lower ID for Pythia versus the other models (as expected) for layers 1-20, but higher ID in later layers?

Clarifying questions:
- On line 107, it is claimed that the study of data manifolds has been explored "in search of the geometrical signatures of generalization." What does this mean? What are "geometrical signatures of generalization", and how are they relevant to this paper?
- Regarding "intrinsic dimension" experiments:
    - What do n1 and n2 denote? (See, e.g., line 142-146)
    - What the "known result for the TWO-NN estimator"? What assumptions are made in obtaining this result, and are they appropriate in this context? How is this known result relevant to the experiments carried out in this work (e.g., how is it "exploited")?
    - Are all ID results presented throughout the paper obtained using GRIDE, or do the authors modify GRIDE in any way (or use procedures developed in other works) to compute ID?
- Regarding the shuffling algorithm:
    - Are the order of blocks being permuted, or is it the order of tokens inside each block? (The experiment only makes sense to me in the second case, but it would be good to explicitly state this.)
        - In the first case, then in line 238-240, what do we mean that "S = 0 corresponds to no shuffling?" Wouldn't the "randomPermutation(blocks)" operation in Alg 1 mean that "S = 0" means that the entire thing is shuffled?
        - In the second case, for how many blocks are the order of tokens in the block permuted? (Presumably all blocks?)
    - All in all, even with the requested clarifications added, it might be helpful to provide an example to better clarify this procedure.
- For all figures where LLAMA is the only model being considered -- i.e., I believe this is everything in sec 4.2? -- this should be explicitly noted; otherwise it is not clear what model is being considered, particularly when jumping between sections.
    - Similarly for "layer 10" in Fig 3, the figure should mention in the caption that it is a "histogram of [...] at the layer with maximum ID (L = 10)", or something similar.
- How should the results in "distribution of tokens at the peak" paragraph be interpreted? What do the findings in Fig 3 or the supporting text mean? This experiment is not explained in sections 3 or 4.2 as the others are. (Possibly, this would be a given if the provided summary of related work in the "intrinsic dimension" paragraph of sec 3 was clearer.)
- In line 404-405, why is "more comprehensive analysis required to confirm this statement"? what kind of analysis is being requested, and why is it necessary?
- In line 489-490, where it is stated that ID peaking at similar layers regardless of inputs being in-distribution or not "raises questions about the connection to the transformers’ architecture and the potential for formulating this phenomenon through analytic approaches" -- what does this mean? What questions are these, and why are they raised by ID peaking at similar layers for OOD inputs?

---

> ### Author Response · Authors · 2024-11-22
> **Specialized Response to Reviewer bby9**
>
> ## Reviewer bby9
> >The primary weakness is that the motivation and contribution of the presented analysis is not clear. At a
> high-level, the general motivation for improving our understanding of LLMs by studying the geometry of
> their embedding spaces is not clear. That is, how would our broader understanding of LLMs or other foundation models be advanced by studying the geometric properties considered in this work? For example, at a
> foundational level, are there questions about how or what any given LLM has learned from its training data
> that could be answered on this basis, or could predictions be made about likely behaviors (outputs)? Or at an
> applied level, are there elements of LLM training or deployment that could be improved given the findings of
> this work?
>
> We have mostly addressed these questions in the general response section: Why do token geometry?
> We just add a couple of targeted responses here. Regarding predictions about likely behaviours (outputs),
> the idea of our paper is that by probing the empirical measure of internal representations using our metrics
> we learn global properties about the probability distribution of next-token predictions. Specifically, in this
> work, we find that ID correlates to loss.
>
> As for understanding what an LLM has learnt from its training data - the results from our experiments suggest that ID increases with both the model expressivity and the perplexity (loss) of the input prompt. We also believe there are other factors that impact the ID profile - for instance the unigram frequency of the tokens. A more precise understanding of how these factors influence the ID profile (and other geometric quantities) can allows us to comment on the questions about what a given LLM learnt from a prompt. We note that it goes beyond the scope of the present paper and reserve it for future work.
>
> Regarding LLM training, we added a discussion about this in the conclusions, and added a supporting analysis on training data from Pythia (see plot in Appendix E) to show the evolution of ID profile with training. While we show a sensitivity of ID to learning, we note that a further analysis is required to improve elements of LLM training and deployment.
>
> > At a lower-level, the specific importance of the studied properties (ID, NO, and cosine similarity) is not clear,
> nor is it clear why LLMs are studied at different levels of noise (shuffling) applied to inputs. There are also
> other more detailed questions about experiments that need to be clarified, as discussed in more detail below...
>
> As explained in the general response, our metrics are observational probes of the empirical measure
> of each layer, which is connected to the probability distribution of the next token. The motivation for using
> these types of probes is mostly inspired prior work on prompt-level studies of internal representations. As for
> the motivation for the shuffling experiment, the idea is to probe the geometry of internal representations for
> out-of-distribution tokens with a twofold advantage: first, it provides a baseline to our metrics estimations,
> i.e. we can make sure that our experiments in-distribution indeed expose some model behavior, and not some
> random noise or estimator artifact. Secondly, it clarifies further the ID-loss correlation. As an added value,
> it allows us to compare to prior work on prompt-level results, which would seem counterintuitive (lower ID
> peak with shuffled input, which is the opposite behaviour to what we see at token level). We dedicate an
> Appendix section (A) to this comparison.

---

> > ### Author Response · Authors · 2024-11-22
> > **(continuation of response)**
> >
> > > Additionally, most of the experiments and results are not contextualized with respect to prior work in this area.
> > That is, with the exception of two experiments – cosine similarity (sec 4.2.1) and ”correlation between ID and
> > loss” (end of sec 4.3) – it is not clear how the design and findings of each experiment relate to those of prior
> > work, hindering the reader’s understanding of the contribution offered by this work and its potential impact
> > on the broader research literature. Finally, most of the broader literature studying the internal representations
> > learned by LLMs is not discussed as related work. Only a narrow sliver of work studying LLM geometry is
> > mentioned – i.e., work studying embeddings as particle systems, or according to ID, NO, or cosine similarity
> > – but there are other broad and longstanding bodies of work (such as probing, circuit discovery, feature
> > attribution, etc.) that also aim to mitigate the ”black box problem” and understand the internals of foundation
> > models such as LLMs. (For a few relevant surveys of such work, see, e.g., https://arxiv.org/abs/2408.05859 or
> > https://arxiv.org/abs/2404.14082 – I will refer to this broad category of work studying the black box problem
> > of foundation model internals as ”mechanistic interpretability”, or MI for short.) While it is not necessary
> > to summarize all such work in detail, and entirely appropriate to focus on the most relevant work studying
> > embedding geometry, it is nonetheless important to position this work in the broader literature working toward
> > understanding LLM representations – otherwise, as noted above, the contribution and broader impact of this
> > work is not clear.
> >
> > We have improved the contextualization of our work both in the introduction and in the related work
> > section.
> >
> > >While some brief thoughts regarding the motivation are laid out in the last paragraph of the conclusion,
> > they are incomplete; and this also needs to be made clear early on in the paper (i.e., in the introduction),
> > as otherwise, on a single forward pass of the paper, it’s impossible to interpret or contextualize the results
> > throughout the paper.
> >
> > We agree that the motivation should be spelled out more clearly, and earlier in the paper. Following
> > what introduced in the general response we changed the text to reflect this suggestion.
> >
> > > I suggest that, at minimum, a brief overview of (1) what is studied in MI, and (2) the relationship between
> > MI and the geometric questions studied here, should be discussed. In the ideal case, it would also be useful
> > to discuss what insights could be provided by the kind of geometric analysis considered in this work that are complementary to MI, that could validate or contradict key findings in MI, etc. E.g., there is currently a
> > hot debate in MI over ”how linear” LLM representations are – see, e.g., https://arxiv.org/abs/2406.01506 and
> > https://arxiv.org/abs/2405.14860. While it is of course not the goal of this paper to address such questions,
> > I am sharing this context to provide a typical example of how questions around embedding geometry are
> > discussed in MI, which might be useful for the authors in contextualizing their work.
> >
> > We thank the referee for suggesting a clear improvement in the presentation of our paper. We believe
> > we have better contextualized our work in the introduction and related work sections. Particularly, the relation
> > to logit and tuned lens discussed in the general response provides an essential building block of our analysis.

---

> > > ### Author Response · Authors · 2024-11-22
> > > **(continuation of response)**
> > >
> > > > I also noticed a few minor errata:
> > > line 183: typo/missing words sec 4.2.3: need to mention which figure is being discussed – it is all in reference
> > > to Figure 4, correct?
> > > line 487-489: incomplete sentence.
> > >
> > > Fixed.
> > >
> > > >Finally, I would like to note that – given that the experimental work carried out in this paper is quite good,
> > > and these weaknesses are primarily a matter of framing, clarity of contribution, providing adequate context,
> > > etc. – I am quite open to raising my rating if the authors are able to address the above weaknesses and answer
> > > the questions provided.
> > >
> > > We thank the referee for being open to re-evaluate their rating.
> > >
> > > > Why is it meaningful to ”progressively disrupt the syntactic and semantic structure” of text by increasing the
> > > amount of token shuffling? What would we expect to happen as text is shuffled? And why do we care about
> > > this, given that there is no reason that one would expect texts to be shuffled before passing them to LLMs?
> > > While there is some discussion on this topic in lines 490-494, it does not answer any of these questions –
> > > that is, while naturally one would expect cosine and ID to change when we shuffle the input, why is this an
> > > interesting/important finding? Why is ”further probing of these distributions and their relation to the model’s
> > > processing dynamics” essential? How do the approaches studying embedding geometry from the perspective
> > > of ”particles in dynamical system” (Geshkovski et al., 2023; Cowsik et al., 2024) relate to this work?
> > >
> > > We refer to the general response for the broad response to this question, especially on the connection
> > > to (Geshkovski et al., 2023; Cowsik et al., 2024). As for the motivation for the shuffling experiment, the idea is to probe the geometry of internal representations for out-of-distribution tokens with a twofold advantage: first, it provides a baseline to our
> > > metrics estimations, i.e. we can make sure that our experiments in-distribution indeed expose some model
> > > behavior, and some random noise or estimator artifact. Secondly, it clarifies further the ID-loss correlation.
> > > As an added value, it allows us to compare to prior work on prompt-level results, which would seem counterintuitive (lower ID peak with shuffled input, which is the opposite behaviour to what we see at token level,
> > > see Appendix A.
> > >
> > > >Why are there 50 prompts maximum for all experiments in sec 4.2 (but not 4.3)? This is an exceptionally
> > > small-sample experiment in context of LLMs and the Pile dataset being sampled from – is the number being
> > > kept so small because one or more of the experiments is particularly computationally expensive? If so, which
> > > ones, and what is the main bottleneck? Do we expect results to look similar for a larger sample? Additionally,
> > > how how are the 50 prompts sampled from the 2252 prompts after filtering (per line 190-191)? If the sampling
> > > is not purely random, then the sampling procedure must be explained and motivated.
> > >
> > > To improve the soundness of our estimation of the variance over prompts, we have rerun the results
> > > using all the prompts. All relevant figures now reflect this change. Results are almost identical to the 50
> > > random prompts case, signaling that the estimation of the variance stabilizes for low number of prompts.
> > >
> > > > A major motivator for experiments in sec 4.3 is that the Pile is used to train Pythia but not Mistral or LLAMA;
> > > but is it actually public information whether the Pile was used to train LLAMA 3 8B or Mistral? This
> > > uncertainty could carry major implications for the findings in sec 4.3. (Per the model papers cited by the
> > > authors in sec 4.1, neither model’s training data is disclosed; and my understanding is that this information
> > > has not been made public since the original model papers, either.) If I am correct and it is not known whether
> > > either model is trained on the Pile, then this should be made clear in the paper (both in sec 4.1 and the
> > > uncertainty this carries for the results throughout sec 4.3); and otherwise, any source disclosing these models’
> > > training data should be cited.
> > >
> > > The referee is absolutely correct: Llama and/or Mistral might have been partly trained on Pile. Since we are not aware of any reference confirming, or rejecting, this option, we have mildened
> > > our statements related to this in the text.
> > >
> > > >In figure 5, why do we see lower ID for Pythia versus the other models (as expected) for layers 1-20, but
> > > higher ID in later layers?
> > >
> > > As a premise, we should notice that the difference in height in this case is not very significant, due to
> > > large variance. Nevertheless, the reason could be again related to ID-loss correlation: since Pythia typically
> > > is a worse model (lower performance) we expect higher ID in later layers.

---

> > > > ### Author Response · Authors · 2024-11-22
> > > > **(continuation of response)**
> > > >
> > > > >On line 107, it is claimed that the study of data manifolds has been explored ”in search of the geometrical
> > > > signatures of generalization.” What does this mean? What are ”geometrical signatures of generalization”, and
> > > > how are they relevant to this paper?
> > > >
> > > > We agree that this expression is too vague. We have removed it from the text.
> > > >
> > > > >Regarding ”intrinsic dimension” experiments: What do n1 and n2 denote? (See, e.g., line 142-146)
> > > >
> > > > We have clarified the text. n1 and n2 represent the rank of neighbours considered in the estimator. n1 = 2 and n2 = 4 would mean we consider the distances to the second and the fourth nearest neighbors to estimate the ID.
> > > >
> > > > >What the ”known result for the TWO-NN estimator”?
> > > >
> > > > The result is the equation right after. We realise the text was not clear enough, we rephrased for better
> > > > clarity.
> > > >
> > > > >What assumptions are made in obtaining this result, and are they appropriate in this context?
> > > >
> > > > The main assumption of the estimator is local homogeneity (Poisson distributed points within a few
> > > > neighbors around a point). We have made this explicit in a footnote in the text.
> > > >
> > > > >How is this known result relevant to the experiments carried out in this work (e.g., how is it ”exploited”)?
> > > >
> > > > We used it in section 4.2.2 in the paragraph ”distribution of tokens at the peak” and specifically in
> > > > figure 4. We have made the reference to the equation more explicit.
> > > >
> > > > >Are all ID results presented throughout the paper obtained using GRIDE, or do the authors modify GRIDE in
> > > > any way (or use procedures developed in other works) to compute ID?
> > > >
> > > > All ID results are based on an unmodified definition of GRIDE.
> > > >
> > > > >Regarding the shuffling algorithm: Are the order of blocks being permuted, or is it the order of tokens inside
> > > > each block? (The experiment only makes sense to me in the second case, but it would be good to explicitly
> > > > state this.) In the first case, then in line 238-240, what do we mean that ”S = 0 corresponds to no shuffling?”
> > > > Wouldn’t the ”randomPermutation(blocks)” operation in Alg 1 mean that ”S = 0” means that the entire thing is
> > > > shuffled? In the second case, for how many blocks are the order of tokens in the block permuted? (Presumably
> > > > all blocks?) All in all, even with the requested clarifications added, it might be helpful to provide an example
> > > > to better clarify this procedure.
> > > >
> > > > Indeed the shuffling procedure was not very intuitive. We have solved this by adding an explanation
> > > > diagram to the pseudocode (new Figure 1).
> > > >
> > > > >For all figures where LLAMA is the only model being considered – i.e., I believe this is everything in sec 4.2?
> > > > – this should be explicitly noted; otherwise it is not clear what model is being considered, particularly when
> > > > jumping between sections. Similarly for ”layer 10” in Fig 3, the figure should mention in the caption that it is
> > > > a ”histogram of [...] at the layer with maximum ID (L = 10)”, or something similar.
> > > >
> > > > We have explicited these suggestions in the relative Figure captions.
> > > >
> > > > >How should the results in ”distribution of tokens at the peak” paragraph be interpreted? What do the findings
> > > > in Fig 3 or the supporting text mean? This experiment is not explained in sections 3 or 4.2 as the others
> > > > are. (Possibly, this would be a given if the provided summary of related work in the ”intrinsic dimension”
> > > > paragraph of sec 3 was clearer).
> > > >
> > > > We agree that this experiment was not explained in sufficient detail. We have added further explanations in the relevant paragraph.
> > > >
> > > > >In line 404-405, why is ”more comprehensive analysis required to confirm this statement”? what kind of
> > > > analysis is being requested, and why is it necessary?
> > > >
> > > > In this case, we mean that we should explore more models/examples where we have training data
> > > > available to make a more concrete statement. We have clarified the text by connecting this to the fact that, as
> > > > the referee points out, we do not know whether Mistral or Llama were partially trained on Pile as well.
> > > >
> > > > >In line 489-490, where it is stated that ID peaking at similar layers regardless of inputs being in-distribution or
> > > > not ”raises questions about the connection to the transformers’ architecture and the potential for formulating
> > > > this phenomenon through analytic approaches” – what does this mean? What questions are these, and why
> > > > are they raised by ID peaking at similar layers for OOD inputs?
> > > >
> > > > We agree this statement is too vague. We had in mind our main motivation of probing the empirical
> > > > measure of hidden layers. We made this reference to our motivation more precise.

---

> > > > > ### Comment · Reviewer_bby9 · 2024-11-27
> > > > > **Response to rebuttal**
> > > > >
> > > > > Thank you for your comprehensive response and excellent revised manuscript! I have read the general response to all reviewers, your rebuttal, and the sections of the revised draft that you referenced in your response. Between these, you have fully addressed most of the weaknesses and questions raised in my original review. I have only two remaining (minor) concerns.
> > > > >
> > > > > ### Minor concerns
> > > > >
> > > > > First, and most importantly: in both the manuscript and your response, you frame the shuffling procedure as pushing input prompts "out-of-distribution" (OOD), but I do not think this is an appropriate characterization. While it is true that shuffling would technically push inputs outside the training distribution of LLMs, this is a very artificial "distribution shift" that does not correspond to any naturally-occurring text distribution, as a sufficient degree of shuffling (with high $S$) will destroy the linguistic structure of inputs and -- to adopt the "manifold hypothesis" framing of the paper -- push these inputs off the natural-language manifold. Instead, in my experience, OOD is typically used in reference to inputs that are still naturally-occurring (i.e., "on the manifold" -- or at least close to it, as for, e.g., adversarial perturbations), but simply happen to fall well outside the distribution that the model was trained on, such as providing a model trained on academic articles with social media posts as inputs. Correspondingly, I especially disagree with the statement in the general response that "The use of shuffled versus non-shuffled inputs was acknowledged as an effective method for understanding out-of-distribution scenarios."
> > > > > - *Suggestion:* I request that the authors adopt a more appropriate framing for the shuffling algorithm and corresponding claims about OOD findings -- e.g., the description in Section 4.2 that shuffling "progressively disrupt[s] the syntactic and semantic structure while preserving unigram frequency distribution" would be reasonable. At minimum, it should be clarified that results obtained using the shuffling algorithm cannot be taken as a reasonable proxy for any "real-world" OOD scenario.
> > > > >
> > > > > Second, the authors "added a supporting analysis on training data from Pythia (see plot in Appendix E) to show the evolution of ID profile with training." I do not know how to interpret the results in this plot -- e.g., what does "K" refer to in the legend, and how do these findings relate to the discussion in Appendix A? No analysis is provided in either the revised manuscript or rebuttal.
> > > > > - *Suggestion:* I do not think these results are essential to the story (particularly as they are never referenced in the main paper). Thus, while it would be ideal to provide clarification and analysis on this point, it would also be an acceptable solution to simply remove this result.
> > > > >
> > > > > ### Assessment
> > > > >
> > > > > These are only minor issues in comparison to my initial concerns about the paper, the rest of which have been fully resolved. Thus, given the substantial improvements in the revised manuscript and what are (now) clear, well-motivated, contextualized, and significant contributions, I will edit my original review to increase my score to **8: accept, good paper**.

---

> > > > > > ### Author Response · Authors · 2024-11-28
> > > > > > **Response to Reviewer bby9**
> > > > > >
> > > > > > We once again thank the reviewer for a very thorough analysis of our work, their feedback has been instrumental to improve our work significantly. We are grateful to the reviewer for recognizing our work and raising the score.
> > > > > >
> > > > > > Here we briefly reply to the reviewers suggestions:
> > > > > >
> > > > > > > Suggestion: I request that the authors adopt a more appropriate framing for the shuffling algorithm and corresponding claims about OOD findings -- e.g., the description in Section 4.2 that shuffling "progressively disrupt[s] the syntactic and semantic structure while preserving unigram frequency distribution" would be reasonable. At minimum, it should be clarified that results obtained using the shuffling algorithm cannot be taken as a reasonable proxy for any "real-world" OOD scenario.
> > > > > >
> > > > > > The reviewer is right on this, we have been imprecise in the use of the "out-of-distribution" expression in this context. We agree with the reviewer's suggestion and modified the paper accordingly, removing instances where needed and rephrasing our claims.
> > > > > >
> > > > > > > Suggestion: I do not think these results are essential to the story (particularly as they are never referenced in the main paper). Thus, while it would be ideal to provide clarification and analysis on this point, it would also be an acceptable solution to simply remove this result.
> > > > > >
> > > > > > Interpretation of the plot in Appendix E: This plot examines how the ID profile changes during training by analyzing the ID profile at various training steps, as indicated by "Checkpoint" in the legend. The "K" in the legend refers to *1000*, for example, 4K refers to the $4000^{\text{th}}$ step in training. We have made this change in the legend to make it clearer. This figure is referred to in the Conclusions section of this paper where we briefly discuss the prospects of analyzing the ID profile during training. While we acknowledge that the figure is not central to our analysis, it provides intuition about the ID profile at different steps of training and helps motivate future research directions. Thus we leave it in the current version, with the appropriate modifications.

---

### Official Review · Reviewer_B7tV · 2024-11-04

**Soundness:** 4
**Presentation:** 3
**Contribution:** 2
**Rating:** 5
**Confidence:** 4

**Summary:**

This work explores the geometry of transformer models using a few metrics (cosine similarity, nearest neighbors overlap between layers, and ID). They also see how these metrics change as the text data becomes more shuffled.

**Strengths:**

The paper is clearly written, and the plots are clear. The experiments seem sound and well-executed. In general, this is an interesting topic– how can we measure the geometry of the model, and how does it relate to the inputs/outputs?

**Weaknesses:**

My main criticism of this work is that it is missing motivation. Like I said, I do think that this is an interesting area. However, I am not sure what the takeaways are, or what one should do with the information that these metrics change in specific ways. Other than the “correlation between ID and loss” experiment, are these observations connected to model behavior or control?

I appreciate the desire to formalize methods mathematically, but I think some of it can be removed for readability (e.g., the definition of cosine similarity)

There is a lot of missing related work analyzing the geometry of transformers, e.g., probing, David Bau’s work, the recent mechanistic interpretability work, etc. I’m not saying you have to go into great detail here, but it would be good to mention that these areas exist and contextualize your work within them.

Nits:
Some of your citations are from ArXiv, but were later published in peer-reviewed venues (e.g., Intrinsic dimension of data representations in deep neural networks was a NeurIPs paper). Consider swapping out references when possible.

“Model transformers” → “transformer models”? (in the related work section titles)

The ID section (and each section for your metrics) could benefit from a diagram to provide some intuition.

The “shuffling method” might be better explained by a diagram than pseudocode.

**Questions:**

“if the model is trained on a dataset that contains the input prompts, then it shows lower ID peaks and higher NO, suggesting better adaptability”--> I was a little confused about this. Adaptability to what?

“The calculation of intrinsic dimension has a long history of successful estimators of various types” → I’m having trouble parsing this. Do you mean “there are various good estimates of ID”?

---

> ### Author Response · Authors · 2024-11-22
> **Specialized response to Reviewer B7tV**
>
> ## Review B7tV
>
> > My main criticism of this work is that it is missing motivation ... Other than the “correlation between ID and
> loss” experiment, are these observations connected to model behavior or control?
>
> This is indeed a fair point which we have tried to clarify in the general response (see above). The
> core message is that we want to explore how token geometry influences next token prediction in transformers
> by examining empirical measures across layers. As an observational probe of the empirical measure, we
> use intrinsic dimension, neighborhood overlap, and cosine similarity among tokens across layers. We have
> made the correlation between ID and loss more explicit by showing it through three steps: ID in internal
> representations correlates with logits ID, logits ID correlates with softmax entropy, which in turn correlates
> with loss. These findings relate explicitly the geometry of tokens to the probability distribution of next token
> prediction. As an outlook to potential applications, we argue that this correlation might be relevant during
> training (see conclusions).
>
> > I appreciate the desire to formalize methods mathematically, but I think some of it can be removed for readability (e.g., the definition of cosine similarity)
>
> We have implemented this suggestion by removing the suggested definition and a few unneeded formal definitions in the methodology section, as indeed it improves readability.
>
> > There is a lot of missing related work analyzing the geometry of transformers, e.g.,probing, David Bau’s
> work, the recent mechanistic interpretability work, etc. I’m not saying you have to go into great detail here,
> but it would be good to mention that these areas exist and contextualize your work within them.
>
> We agree that the related work section was missing relevant literature in the domain addressed by this
> paper. We have improved this section.
>
> > Some of your citations are from ArXiv, but were later published in peer-reviewed venues (e.g., Intrinsic
> dimension of data representations in deep neural networks was a NeurIPs paper). Consider swapping out
> references when possible.
>
> We thank the referee for pointing this out, and we have made the necessary corrections wherever applicable.
>
> > “Model transformers” → “transformer models”? (in the related work section titles)
>
> We fixed the typo.
>
> >The ID section (and each section for your metrics) could benefit from a diagram to provide some intuition.
>
> While we recognize the added value of diagrams for intuition, because of the page limit, we decided
> not to add one for each metric, as suggested by the referee.
>
> We did add a diagram to explain the shuffling
> method (see below).
>
> > The “shuffling method” might be better explained by a diagram than pseudocode.
>
> We thank the referee for the suggestion. We added a schematic diagram to explain the shuffling
> method more explicitly (Figure 1 of revised version).
>
>
> ### Questions
>
> > “if the model is trained on a dataset that contains the input prompts, then it shows lower ID peaks and higher
> NO, suggesting better adaptability”→ I was a little confused about this. Adaptability to what?
>
> We agree with the referee that this sentence needed rephrasing. We actually removed the entire
> sentence from the current version.
>
> > “The calculation of intrinsic dimension has a long history of successful estimators of various types” → I’m
> having trouble parsing this. Do you mean “there are various good estimates of ID”?
>
> We have rephrased the sentence in the text. Here we meant that indeed there is a significant number
> of works developing accurate estimators for the intrinsic dimension of manifolds.

---

### Author Response · Authors · 2024-11-22
**General Response To Reviewers**

# General response
We thank the reviewers for a thorough reading of our work and for providing constructive feedback on our weaknesses. Given that the reviews share key points of criticism, we are addressing them collectively here, to facilitate
reading. We also provide specialized replies in each thread to integrate this general response.

**Strengths**
- We appreciate the reviewers’ recognition of the soundness and clarity of our experiments and methodology.
- Our approach of analyzing token representations rather than entire prompts has been recognized as offering
a finer-grained perspective that could lead to meaningful insights about transformer dynamics.
- The use of shuffled versus non-shuffled inputs was acknowledged as an effective method for understanding out-of-distribution scenarios.

**Weaknesses**. The primary concern raised by reviewers was the lack of clear motivation and contribution to the
field of LLMs interpretability. Related, it was unclear to some reviewers the impact on model behavior of our findings. The discussion of related work was considered insufficiently contextualized within mechanistic interpretability and geometric approaches.

## Why do token geometry?
The central motivation of this work is the study of how token geometry connects to next token prediction. An
important aspect of this connection uses the notion of empirical measure, analyzed in
[1, 2] in the context of transformers. Given $n$ points at positions $x_1, \ldots, x_n \in \mathbb{R}^d$ (a point cloud), their empirical measure is the probability measure $\mu=\frac{1}{n} \sum_{j=1}^n \delta_{x_j}$, i.e., the empirical measure encodes the distribution of points in the embedding space.
Thus, given a sequence of tokens (prompt) embedded in $\mathbb{R}^d$ in the embedding layer, we have a point cloud, and thus an empirical measure, for every layer. The empirical measure for the last layer is the output measure.

Transformers can be understood as an evolving mean-field interacting particle system [1], where the evolution of token representations is influenced by their empirical measure [2]. This emphasizes the need to understand the empirical measure at each layer. The mean field interacting particle picture was used to show that tokens tend to form clusters [18] when the weights are not time dependent. This clustering behaviour can be associated with the empirically observed rank collapse phenomenon in transformers [3-7].

An important insight from [1] in the context of next token prediction is that the output measure of tokens encodes the probability distribution of the next token, and its clustering indicates a small number of possible outcomes. A complementary perspective to the evolution of token representations across layers can be gained by studying the latent predictions of transformer models [17] from the framework of iterative inference [8] which indicates that the probabilities of the next tokens are incrementally updated layer by layer. [9] suggests that causal LLMs develop a reasonably accurate prediction regarding the next token in the middle layers, with subsequent layers refining
these predictions. This means we should expect the empirical measures of internal layers to
reflect this trend, i.e. a rapid change of the empirical measure in the early layers and a more refined
change towards the later layers. Since the latent predictions are obtained by unembedding the residual stream [19] and our methods understand the geometric properties of the residual stream, we can expect the statistical properties (eg. entropy) of the latent prediction probabilities to be encoded in the geometry of the internal representations of the tokens.

These findings motivate us to examine the empirical measure from a geometric perspective. To observationally probe it, we draw inspiration from previous works using intrinsic dimension and neighbourhood overlap to study the geometry of internal representations [10-16]. In these works, an important difference is that point clouds are built as a collection of prompts represented as a single point
(the last token), not from the full sequence of tokens in a prompt, thereby lacking the direct link to the empirical measure. Additionally, we calculate cosine similarity as a general probe of pairwise relations among tokens.

In summary, token geometry characterizes the empirical measure that is primarily responsible for the evolution of token representations in a mean-field interacting particle system approach. In the context of next token
prediction, studying the empirical measure is useful since the output measure encodes information about the next
token probabilities.
We use intrinsic dimension and cosine similarity as observables that probe the structural
properties of the empirical measure. To understand the effect of the empirical measure on the dynamics of the
token representations, we use neighborhood overlap to track the neighborhood composition
between adjacent layers.

---

> ### Author Response · Authors · 2024-11-22
> **(continuing general response)**
>
> ## Relation to prompt-level studies
> Previous work [10- 16] has studied internal representations from a geometric point of view by
> considering point clouds of last token representations as observable. While the approach is similar in spirit, token-level and prompt-level measures of intrinsic dimension, neighbor overlap, and cosine similarity probe different
> manifolds and thus different features of LLMs.
>
> While prompt-level and token-level ID profiles exhibit similar behavior qualitatively, e.g.
> they peak in early-middle layers, there is a notable difference in the shuffled and unshuffled prompts. At the
> prompt level, we see that the unshuffled ID has a more prominent peak than the shuffled ID, whereas it is the
> other way around at the token level. This difference between token and prompt level ID curves offers a window to understand the difference
> between the token and prompt geometries. The core reason for this diverging behavior is that we are looking at
> different manifolds and thus observing two distinct behaviors: token level ID is correlated to the input perplexity
> (measured using the cross-entropy loss) and the prompt level ID is a measure of the semantic information [13, 15]. Given a dataset
> of the prompts with a high perplexity at the token level such as in the shuffled case, we can expect the last token
> representations to be less likely to share semantic content, leading to a lower intrinsic dimension at the prompt level.
> At the token level, the lesser prominence of the peak of the unshuffled case can be explained using the ID loss correlation. Since the loss is expected to be lower for the unshuffled prompts, we can expect their ID peak to be less prominent than that of the shuffled prompts.
>
> ## Relevance and impact of model behavior
> The main result of our experiments is that we find a statistical relation between the geometry of tokens and the
> probability distribution of the next token: the intrinsic dimension of the token representations across hidden layers
> is correlated to the average cross-entropy loss of the next token probability distribution for a given prompt. This
> suggests that prompts with a higher cross-entropy loss have token representations lying in higher dimensional
> manifolds. While the ID-loss correlation was already anticipated in the original submission, we have made the
> argument stronger and contextualized it better by providing more evidence: we elaborate the relation through three
> steps, connecting ID of internal representations to ID of logits, which in turns correlates with softmax entropy,
> which is related to the model’s loss. We provide details on these steps in Appendix D. In this appendix, we also
> provide an explicit analytic calculation of this correlation within a simplified scenario, which suggests that the
> relation is entropy $\propto \log ID$. We believe that this result confirms that our approach is valid, and brings further
> insights on model behavior.
>
> This correlation is certainly relevant during training. As already shown at the prompt level in prior work [15],
> ID is mostly constant and low at the early stages of training, but it increases as training progresses. At the token level, we observe that the
> ID tends to rise due to enhanced model expressivity in the early stages, while there is also a tendency for ID to decline as the minimization of loss improves towards the later stages of training. We think it might be interesting to investigate these aspects in more detail, but we leave this to future work.
>
> [changes in text] Edited summary of results in the introduction, expanded section about ID-loss in the main
> body, expanded conclusions, and added Appendix D with the theoretical calculation of the relation between logits ID and
> cross-entropy loss, Appendix E with token level ID profile over training for Pythia.

---

> > ### Author Response · Authors · 2024-11-22
> > **(references)**
> >
> > # References
> >
> > [1] B. Geshkovski, C. Letrouit, Y. Polyanskiy, and P. Rigollet, “A mathematical perspective on transformers,”
> > 2024. https://arxiv.org/abs/2312.10794.
> >
> > [2] A. Agrachev and C. Letrouit, “Generic controllability of equivariant systems and applications to particle
> > systems and neural networks,” 2024. https://arxiv.org/abs/2404.08289.
> >
> > [3] S. Anagnostidis, L. Biggio, L. Noci, A. Orvieto, S. P. Singh, and A. Lucchi, “Signal propagation in
> > transformers: Theoretical perspectives and the role of rank collapse,” in Advances in Neural Information
> > Processing Systems, 2022.
> > https://openreview.net/forum?id=FxVH7iToXS.
> >
> > [4] H. Shi, J. GAO, H. Xu, X. Liang, Z. Li, L. Kong, S. M. S. Lee, and J. Kwok, “Revisiting over-smoothing in
> > BERT from the perspective of graph,” in International Conference on Learning Representations. 2022.
> > https://openreview.net/forum?id=dUV91uaXm3.
> >
> > [5] X. Wu, A. Ajorlou, Z. Wu, and A. Jadbabaie, “Demystifying oversmoothing in attention-based graph neural
> > networks,” in Thirty-seventh Conference on Neural Information Processing Systems. 2023.
> > https://openreview.net/forum?id=Kg65qieiuB.
> >
> > [6] B. He, J. Martens, G. Zhang, A. Botev, A. Brock, S. L. Smith, and Y. W. Teh, “Deep transformers without
> > shortcuts: Modifying self-attention for faithful signal propagation,” in The Eleventh International
> > Conference on Learning Representations. 2023.
> > https://openreview.net/forum?id=NPrsUQgMjKK.
> >
> > [7] X. Wu, A. Ajorlou, Y. Wang, S. Jegelka, and A. Jadbabaie, “On the role of attention masks and layernorm
> > in transformers,” in The Thirty-eighth Annual Conference on Neural Information Processing Systems. 2024.
> > https://openreview.net/forum?id=lIH6oCdppg.
> >
> > [8] S. Jastrzebski, D. Arpit, N. Ballas, V. Verma, T. Che, and Y. Bengio, “Residual connections encourage
> > iterative inference,” in International Conference on Learning Representations. 2018.
> > https://openreview.net/forum?id=SJa9iHgAZ.
> >
> > [9] nostalgebraist, “interpreting gpt: the logit lens,” LessWrong (2020) . https://www.lesswrong.com/posts/AcKRB8wDpdaN6v6ru/interpreting-gpt-the-logit-lens.
> >
> > [10] A. Ansuini, A. Laio, J. H. Macke, and D. Zoccolan, “Intrinsic dimension of data representations in deep
> > neural networks,” in Proceedings of the 33rd International Conference on Neural Information Processing
> > Systems. Curran Associates Inc., Red Hook, NY, USA, 2019.
> >
> > [11] D. Doimo, A. Glielmo, A. Ansuini, and A. Laio, “Hierarchical nucleation in deep neural networks,” in
> > Advances in Neural Information Processing Systems, 2020.
> >
> > [12] P. Pope, C. Zhu, A. Abdelkader, M. Goldblum, and T. Goldstein, “The intrinsic dimension of images and its
> > impact on learning,” in International Conference on Learning Representations. 2021.
> > https://openreview.net/forum?id=XJk19XzGq2J.
> >
> > [13] L. Valeriani, D. Doimo, F. Cuturello, A. Laio, A. Ansuini, and A. Cazzaniga, “The geometry of hidden
> > representations of large transformer models,” in Advances in Neural Information Processing Systems,
> > 2023. https://proceedings.neurips.cc/paper_files/paper/2023/file/a0e66093d7168b40246af1cddc025daa-Paper-Conference.pdf.
> >
> > [14] E. Cheng, C. Kervadec, and M. Baroni, “Bridging information-theoretic and geometric compression in
> > language models,” in Proceedings of the 2023 Conference on Empirical Methods in Natural Language
> > Processing, Association for Computational
> > Linguistics, Singapore, Dec., 2023. https://aclanthology.org/2023.emnlp-main.762.
> >
> > [15] E. Cheng, D. Doimo, C. Kervadec, I. Macocco, J. Yu, A. Laio, and M. Baroni, “Emergence of a
> > high-dimensional abstraction phase in language transformers,” 2024.
> > https://arxiv.org/abs/2405.15471.
> >
> > [16] E. Cheng and R. J. Antonello, “Evidence from fmri supports a two-phase abstraction process in language
> > models,” 2024. https://arxiv.org/abs/2409.05771.
> >
> > [17] N. Belrose, Z. Furman, L. Smith, D. Halawi, I. Ostrovsky, L. McKinney, S. Biderman, and J. Steinhardt,
> > “Eliciting latent predictions from transformers with the tuned lens,” 2023.
> > https://arxiv.org/abs/2303.08112.
> >
> > [18] B. Geshkovski, C. Letrouit, Y. Polyanskiy, and P. Rigollet,
> > “The emergence of clusters in self-attention dynamics,” in The Thirty-seventh Annual Conference on Neural Information Processing Systems, 2023. Available: https://openreview.net/forum?id=aMjaEkkXJx
> >
> > [19] N. Elhage, N. Nanda, C. Olsson, T. Henighan, N. Joseph, B. Mann, A. Askell, Y. Bai, A. Chen, T. Conerly, N. DasSarma, D. Drain, D. Ganguli, Z. Hatfield-Dodds, D. Hernandez, A. Jones, J. Kernion, L. Lovitt, K. Ndousse, D. Amodei, T. Brown, J. Clark, J. Kaplan, S. McCandlish, and C. Olah, “A mathematical framework for transformer circuits,” Transformer Circuits Thread, 2021. Available: https://transformer-circuits.pub/2021/framework/index.html.

---

### Meta-Review · Program_Chairs · 2024-12-24

**Metareview:**

PC is entering the meta-review on behalf of SAC and AC:

The major concern noted by reviewers was that the paper made a relatively small advance over existing work, and that there is limited utility in their findings.

**Additional Comments On Reviewer Discussion:**

TBD

---

> ### Public Comment · ~Karthik_Viswanathan2 · 2025-02-20
> **Additional information for the Meta Review**
>
> Dear Program Chairs,
>
> Could you please share more information regarding the meta-review? Any additional details from the reviewer discussion or insights on how I can improve this work would be greatly appreciated.

---

### Decision · Program_Chairs · 2025-01-22

Reject